# Dietary Cocoa Flavanols Enhance Mitochondrial Function in Skeletal Muscle and Modify Whole-Body Metabolism in Healthy Mice

**DOI:** 10.3390/nu13103466

**Published:** 2021-09-29

**Authors:** Frédéric Nicolas Daussin, Alexane Cuillerier, Julianne Touron, Samir Bensaid, Bruno Melo, Ali Al Rewashdy, Goutham Vasam, Keir J. Menzies, Mary-Ellen Harper, Elsa Heyman, Yan Burelle

**Affiliations:** 1ULR 7369—URePSSS—Unité de Recherche Pluridisciplinaire Sport Santé Société, University Lille, University Artois, University Littoral Côte d’Opale, F-59000 Lille, France; samir.bensaid@univ-lille.fr (S.B.); elsa.heyman@univ-lille.fr (E.H.); 2Interdisciplinary School of Health Sciences and Faculty of Health Sciences, University of Ottawa, Ottawa, ON K1H 8M5, Canada; acuil042@uottawa.ca (A.C.); aalre094@uottawa.ca (A.A.R.); gvasam@uottawa.ca (G.V.); kmenzies@uottawa.ca (K.J.M.); yburell2@uottawa.ca (Y.B.); 3INRAE, UMR1019, Unité de Nutrition Humaine (UNH), Équipe ASMS, Université Clermont Auvergne, 63001 Clermont-Ferrand, France; julianne.touron@gmail.com; 4Department of Physical Education, Exercise Physiology Laboratory, Federal University of Minas Gerais, Belo Horizonte, MG 31270-901, Brazil; brunomelo-89@hotmail.com; 5Department of Biochemistry Microbiology and Immunology, Faculty of Medicine, Ottawa Institute of Systems Biology, University of Ottawa, Ottawa, ON K1H 8M5, Canada; MaryEllen.Harper@uottawa.ca

**Keywords:** cocoa flavanols, NAD metabolism, mitochondrial mass, glucose metabolism, skeletal muscle

## Abstract

Mitochondrial dysfunction is widely reported in various diseases and contributes to their pathogenesis. We assessed the effect of cocoa flavanols supplementation on mitochondrial function and whole metabolism, and we explored whether the mitochondrial deacetylase sirtuin-3 (Sirt3) is involved or not. We explored the effects of 15 days of CF supplementation in wild type and Sirt3^-/-^ mice. Whole-body metabolism was assessed by indirect calorimetry, and an oral glucose tolerance test was performed to assess glucose metabolism. Mitochondrial respiratory function was assessed in permeabilised fibres and the pyridine nucleotides content (NAD^+^ and NADH) were quantified. In the wild type, CF supplementation significantly modified whole-body metabolism by promoting carbohydrate use and improved glucose tolerance. CF supplementation induced a significant increase of mitochondrial mass, while significant qualitative adaptation occurred to maintain H_2_O_2_ production and cellular oxidative stress. CF supplementation induced a significant increase in NAD^+^ and NADH content. All the effects mentioned above were blunted in Sirt3^-/-^ mice. Collectively, CF supplementation boosted the NAD metabolism that stimulates sirtuins metabolism and improved mitochondrial function, which likely contributed to the observed whole-body metabolism adaptation, with a greater ability to use carbohydrates, at least partially through Sirt3.

## 1. Introduction

Mitochondria are subcellular organelles that are involved in multiple cellular functions, such as energy transduction through mitochondrial oxidative phosphorylation and in mitochondrial hydrogen peroxide production or mitochondrial-mediated cell death activation [1,2,3]. Emerging evidence suggests that mitochondrial dysfunction is involved in many pathologies and plays a central role in their pathogenesis [4]. Our group reported mitochondrial impairment in patients with type 1 diabetes long before clinical complications [5]. Moreover, a decrease of mitochondrial oxidative capacity occurs with aging even in healthy people [6]. Therefore, identifications of strategies to mitigate these dysfunctions or enhance mitochondrial mass are opportunities to prevent, or at least to reduce, the incidence of many disorders or increase aerobic capacity of healthy subjects.

In this context, the identification of safe and natural compounds that enhance mitochondrial function and whole-body metabolism with or without limited side effects is of interest. Among natural compounds, cocoa flavanols (CF) are considered promising molecules as their consumption was shown to positively affect cardiovascular health, insulin resistance, or immune function [7]. Indeed, CF was shown to improve glucose metabolism, and a short term administration of CF is followed by an increase in insulin sensitivity, even in glucose-intolerant hypertensive patients [8,9].

Among the flavanols, (-)-epicatechin (EPI) is the most commonly found monomer in CF, and EPI is considered the major bioavailable and bioactive molecule of CF [10]. A consistent body of evidence indicates that EPI supplementation improves mitochondrial function and/or content in skeletal muscle [11]. EPI supplementation stimulates multiple pathways converging on peroxisome proliferator-activated receptor gamma coactivator 1-alpha (PGC1α) and major nuclear transcriptional complexes. EPI directly enhances nitric oxide generation, which stimulates mitochondrial biogenesis [11]. Concomitantly, the sirtuins 1 and 3 are also stimulated by EPI supplementation [11]. The use of proanthocyanidins, oligomeric flavonoids which contain EPI, was shown to increase the intracellular nicotinamide adenine dinucleotide (NAD^+^) levels through the increase of several precursors for NAD biosynthesis and upregulate sirtuin-1 mRNA levels in rat liver [12]. The sirtuin family (Sirt), an NAD^+^-dependent deacetylase, comprises seven members that differ by their subcellular distribution, substrate specificity, and cellular functions [13]. Sirt1, an extensively studied member of this family, stimulate mitochondrial biogenesis by promoting deacetylation of PGC1α, thereby enhancing its transcriptional activity [14]. Sirt1, a key metabolic sensor, was stimulated following EPI administration [12,15,16,17]. However, the influence of EPI on other sirtuins have not been explored. The assessment of EPI effect will be particularly interesting on sirtuins located within the mitochondria (Sirt3, Sirt 4, and Sirt5), which are known to modulate the activity of the Krebs cycle and respiratory chain enzymes [18]. Sirt3, which is highly expressed in tissues with high metabolic turnover and mitochondrial content, is of particular interest concerning its critical role in maintaining normal mitochondrial function through reversible protein lysine deacetylation [19,20,21]. Moreover, Sirt3 plays an important role in the regulation of whole-body metabolism [20]. Sirt3 was shown to be altered in skeletal muscle of models of type 1 and type 2 diabetes and cardiovascular diseases [22,23]. Furthermore, it was suggested that activation of Sirt3 might represent a promising therapeutic strategy for the improving mitochondrial function and metabolism [23,24]. Therefore, determining whether Sirt3 underlies part of the beneficial effects of CF supplementation on whole-body metabolism and mitochondrial function is of interest.

In this study, we used an integrative approach to investigate, in mice, the effect of a 15-day CF supplementation. We hypothesised that the CF supplementation would: (i) modify whole-body metabolism and glucose metabolism, (ii) increase mitochondrial function in oxidative and glycolytic muscle in permeabilized fibers, and (iii) boost NAD metabolism. We also explored the putative involvement of Sirt3 in CF-induced mitochondrial biogenesis and whole-body metabolism. Our hypotheses were that CF’s effects on whole-body metabolism and mitochondrial mass would be blunted in Sirt3^-/-^ mice.

## 2. Methods

### 2.1. Animals and Diet

For this study, 129S1/SvlmJ 10-week-old male mice were purchased from Jackson Laboratories (Bar Harbor, ME, USA). To determine whether sirtuins are involved in CF administration-induced mitochondrial mass and function adaptation, we also used 10-week-old Sirt3 whole-body knock-out male mice on a same strain as wild-type [25]. The Sirt3^-/-^ mice were used to explore the involvement of Sirt3 in CF-induced mitochondrial biogenesis and whole-body metabolism adaptations. The mice were kept in a room at a regulated temperature of 23–25 °C and controlled lighting (12-h light and dark cycles). The animals had access to rodent chow and tap water *ad libitum*.

After at least a week of habituation in the animal facility, experimental animals received CF supplementation through oral gavage of a natural extract (302.1 mg/kg body weight twice a day for 15 days) resuspended in carboxymethylcellulose (Sigma-Aldrich, St. Louis, Missouri). The composition of the cocoa powder is described in Table 1 and has been obtained from Naturex (Quart de Poblet, Spain). The composition has been established by high performance liquid chromatography method and 100 mg of extract are equivalent to an average of 475 mg of dry cocoas seeds. The control mice received a vehicle composed of similar content of theobromine and caffeine dissolved in carboxymethylcellulose by oral gavage. The daily dose was determined according to the industry’s guidance that converts the human dose to animal equivalent doses based on body surface area, and we multiplied the human dose by 12.3 [26]. The oral gavage was performed by an experienced technician.

### 2.2. Tissue Collection

Twenty-four hours after the last oral gavage, unfasted animals were anaesthetised using ketamine and xylazine (respectively: 100 and 10 mg/kg). Collection of tissues was performed immediately after anaesthesia was achieved. The soleus and the white portion of gastrocnemius were excised. Samples were either immediately frozen in liquid nitrogen and stored at −80 °C for subsequent analysis or used to prepare skinned muscle fibres.

### 2.3. Metabolic and Physical Activity Assessments

Indirect calorimetry and spontaneous physical activity were measured using the Comprehensive Laboratory Animal Monitoring System (CLAMS; Columbus Instruments, Columbus, OH, USA). Mice were housed individually in the chambers at 28 °C, with light from 07h00 to 19h00 and *ad libitum* access to rodent chow and tap water. Mice were acclimated to the metabolic cages for three days before day one of data collection. The data collection was performed on the fifteenth day of supplementation. Oxygen consumption (VO_2_), carbon dioxide production (VCO_2_), and respiratory exchange ratio (RER) were measured with an airflow of 0.5 L/min and a sample air of 0.4 L/min. Percent relative cumulative frequency (PCRF) of RER was determined over a 24-h period as previously described by Riachi et al. (2004). Briefly, PCRF was determined by the addition of the frequency of each data point to the previous increment sequence (from 0.65 to 1.20 with an increment of 0.01) [27]. The metabolic flexibility was determined by the Hill slope (H value) and the 50th percentile (EC50) of the PRCF curve and by measuring the RER amplitude over 24 h (Riachi et al. 2004). General physical activity was monitored using an infrared beam-operated sensor system that detected *x*- and *z*-axis activity. Both total counts (every time a beam is broken) and ambulatory counts (every time a new beam is broken) were assessed. An X-total (X-TOT) count is registered when the animal moves horizontally more than 0.5 in and is representative of small-scale, repetitive activities, such as scratching and grooming. The X-ambulatory count (X-AMB) measures actual locomotion by registering a count only when the animal breaks a new beam. A Z-total (Z-TOT) count is registered when 1.5 in of vertical movement occurs, i.e., rearing (Abreu-Vieira 2016). For estimating carbohydrate and fat oxidation from VO_2_ and VCO_2_, we used the nonprotein respiratory quotient table from Péronnet and Massicotte previously used in mice [28,29].

### 2.4. Total Body and Fat Mass

Total fat mass and total lean mass were determined in unanaesthetised mice by quantitative resonance interference analysis using the EchoMRI whole-body magnetic resonance analyser (Echo Medical System, Houston, TX, USA). Total fat mass represented the sum of all fat in the body. Total lean mass included mainly muscle and inner organs. Skeletal muscle mass is known to account for the largest portion of total lean mass.

### 2.5. Oral Glucose Tolerance Test (OGTT)

Glycemia was measured using a drop of blood collected from the tail with a glucometer (Accu-Chek Performa, Roche, Argentina). OGTT was performed on mice that were fasted for 5 to 6 h. The measurement was performed on the last day of CF supplementation. After measurement of the basal glycemia (time 0), mice were then injected *per os* with a solution of 20% glucose in sterile water at a dose of 1.5 g glucose·kg body weight^−1^. Glycemia was measured at 15, 30, 60, 90, and 120 min after injection of the glucose solution. The incremental area of the glucose curve was then calculated as a measure of glucose tolerance.

### 2.6. qRT-PCR Analysis

The forward (F) and reverse (R) primers of the selected genes used in this study are presented in Table 2. Total RNA was obtained from pulverised frozen white gastrocnemius muscle using Trizol reagent (Invitrogen Life Technologies, Rockville, MD, USA). The purity, integrity and quantity of RNA were evaluated using a microplate reader (Synergy H1 (Biotek Instruments, Winooski, Vermont). The sample was considered pure when the ratio OD260/OD280 was >1.8. cDNA was synthesised from total RNA (2 µL) using the iScript reverse transcription supermix for RT-qPCR (Biorad, Hercules, CA, USA). Total RT products were analysed by quantitative polymerase chain reaction (PCR) using the 2x Platinium SYBR Green qPCR SuperMix-UDG according to the manufacturer’s specifications (Invitrogen Lige Technologies, Burlington, Canada). Cycling was achieved in a CFX96 cycler (Biorad, Hercules, California; conditions: 95 °C for 10 min and 40 cycles of 95 °C for 30 s, 55 °C for 45 s and 72 °C for 45 s). At the end of each run, the absence of primer-dimer formation and the presence of a unique amplicon were confirmed using the melting curve. The threshold cycle (Ct) values were used following the 2-∆∆Ct method to analyse qPCR data, with either ß-Actin or HPRT1 as a housekeeping gene [30]. Quantitative RT-PCR values were related to those of the control group, which were set to 1.

### 2.7. Enzyme Activities

The measurement of the specific activity of electron transport chain complexes and citrate synthase were performed spectrophotometrically as previously described [31]. Pulverised frozen white gastrocnemius muscles (~30 mg) were homogenised with a vibrating microbead homogeniser in 500 µL of homogenisation buffer (120 mM KCl, 20 mM of HEPES, 2 mM of MgCl2, 1 mM of EGTA, and 5 mg·mL^−1^ of BSA, pH 7.4) followed by the addition of 500 µL of hypotonic media (25 mM potassium phosphate and 5 mM MgCl_2_, pH 7.2). The samples were then submitted to three rounds of freeze-thaw cycle in liquid nitrogen. The samples were centrifuged for 10 min at 600× *g* at 4 °C, and the supernatant was kept on ice until the analysis. The protein content was determined in the supernatant in triplicate using a Pierce BCA Protein Assay Kit (Thermo Scientific, Rockford, IL, USA).

The activity of citrate synthase (CS) was determined spectrophotometrically at 412 nm following the reduction of 2 mM 5,5′-dithio-bis(2-nitrobenzoic acid) in the presence of 0.1 mM acetyl-CoA and 12 mM of oxaloacetic acid in 200 mM Tris buffer (pH 7.4). The rotenone-sensitive NADH-decylubiquinone oxidoreductase (complex I) assay was performed at 340 nm using the acceptor 2,3-dimethoxy-5-methyl-6-n-decyl-1,4-benzoquinone (80 µM) and NADH as electron donor (200 µM) in 10 mM Tris buffer (pH 8.0). The addition of 4 µM of rotenone was used to quantify the rotenone sensitive activity. The activity of the cytochrome c oxidase (complex IV) was performed at 550 nm using 10 µM reduced cytochrome c as donor and 2.5 mM of n-dodecyl-ß-maltoside to permeabilise both mitochondrial membranes in 100 mM potassium phosphate buffer (pH 7.0).

All the enzyme assays were determined in triplicate, and results were related to those of the control group, which were set to 100.

### 2.8. Mitochondrial Assessment

Mitochondrial respiration and mitochondrial H_2_O_2_ production were studied in situ in saponin permeabilised fibres using white gastrocnemius and soleus muscles [1]. Briefly, fibres were separated under binocular microscope in solution A at 4 °C (in mM: 2.77 CaK_2_ EGTA, 7.23 K_2_EGTA, 6.56 MgCl_2_, 20 taurine, 0.5 DTT, 50 K-methane sulfonate, 20 imidazole, 5.7 Na2 ATP, and 15 creatine-phosphate, pH 7.1) and permeabilised in solution A with 50 µg·mL^−1^ of saponin for 30 min at 4 °C. Mitochondrial respiratory function was determined in an oximeter equipped with a Clark-type electrode (Oxygraph, Hansatech Instruments, Glasgow). The chamber was filled with 1 mL of Solution B (in mM: 2.77 CaK_2_ EGTA, 7.23 K_2_EGTA, 6.56 MgCl_2_, 20 taurine, 0.5 DTT, 50 K-methane sulfonate, and 20 imidazole, pH 7.1) and after recording baseline oxygen content in the chamber, one bundle of 1–2 mg dry weight of permeabilised myofibres was placed into the chamber, which was then sealed shut. After baseline readings, the following additions were sequentially made: palmitoyl carnitine and malate (160 µM:5 mM), glutamate (10 mM), succinate (25 mM), rotenone (0.5 µm), the uncoupler CCCP (1 µM), antimycin-A (8 µM), and N,N’,N’-tetramethyl-p-phenylenediamine dihydrochloride and ascorbate (0.9:9 mM). Respiration was measured at 23 °C under continuous stirring. At the end of each test, fibres were carefully removed from the oxygraphic cell, blotted and dried at least 24 h at ~80 °C for determination of fibre weight. Rates of O_2_ consumption (JO_2_) were expressed in nmol O_2_·min^−1^·(mg dry weight)^−1^. All measurements were performed at least in duplicate.

Net H_2_O_2_ release by respiring mitochondria was measured in permeabilised fibre bundles with the fluorescent probe Amplex Red (20 μM: excitation-emission: 563–587 nm), as described previously [32]. Following preparation of permeabilised fibres, samples destined for H_2_O_2_ measurements were rinsed three times in buffer Z (in mM: 110 K-Mes, 35 KCl, 1 EGTA, 5 K_2_HPO_4_, 3 MgCl_2_6H_2_O, and 0.5 mg⋅mL^−1^ BSA, pH 7.3 at 4 °C). Fibre bundles (0.3–1.0 mg dry weight) were incubated at 37 °C in a quartz microcuvette with continuous magnetic stirring in 600 μL of buffer Z (pH 7.3 at 37 °C) supplemented with 1.2 U⋅mL^−1^ horseradish peroxidase. Baseline fluorescence readings were taken in the absence of any exogenous respiratory substrates. The following additions were then made sequentially: glutamate (5 mM), succinate (5 mM), rotenone (0.5 µM), ADP (10 mM), and antimycin-A (8 μM). At the end of each test, fibres were carefully removed from the cuvette, blotted, and dried to determine fibre weight. Rates of H_2_O_2_ production were expressed in AU·min^−1^·(mg dry weight)^−1^ and per oxphos capacity which corresponded to the maximal mitochondrial respiration under phosphorylation state. All measurements were performed at least in duplicate.

Mitochondrial calcium retention capacity (CRC) was assessed in white gastrocnemius muscle ghost fibres as previously described [33]. Briefly, fibres were incubated in a quartz microcuvette under continuous stirring in 600 µL of CRC buffer (in mM: 250 sucrose, 10 MOPS, 0.005 EGTA, and 10 P_i_-Tris, pH 7.3). Fibres were then exposed to a single pulse of 20 nM of Ca^2+^. Changes in extramitochondrial calcium concentration were monitored fluorometrically using Calcium green 5N (1 mM: excitation-emission: 505–535 nm). Permeability transition pore susceptibility (PTP) was assessed by measuring the time required for PTP opening, and CRC was taken as the total amount of Ca^2+^ accumulated by mitochondria before its release. At the end of each test, fibres were carefully removed from the cuvette, blotted, and dried to determine fibre weight. CRC values were expressed in nM of Ca^2+^ per mg of dry weight.

### 2.9. Western Blot Analysis

Proteins from white gastrocnemius and soleus muscles were diluted in Laemmli buffer and separated on Mini-PROTEAN TGX Stain-Free 10% precast polyacrylamide gels (Biorad); an internal standard was loaded on each gel. Electrophoretic separation was done at 200 V for 35 min in migration buffer (25 mM TrisBase, 0.2 M glycine, and 1% SDS (p/v)). Stain-Free (SF) technology contains a proprietary trihalo compound that reacts with proteins, rendering them detectable through UV exposure. SF imaging was performed using ChemiDoc MP Imager and Image Lab 4.0.1 software (Biorad, Hercules, California, USA) with a 5-min stain activation time, and total protein patterns were therefore visualised. Proteins were then transferred on a 0.2 μm nitrocellulose sheet using the Trans-Blot Turbo Transfer System (Biorad). The quality of transfer was controlled by imaging membranes using the SF technology. Following the transfer step, the carbonyl proteins were derivatised with dinitrophenylhydrazine diluting in 2N HCL, and were finally extensively washed with methanol. The membranes were blocked with 5% non-fat dry milk in Tris-buffered saline containing Tween-20 (TBST: 15 mM Tris/HCl, pH 7.6, 140 mM NaCl, and 0.05% Tween-20) for 1 h at room temperature. Membranes were then incubated at 4 °C overnight or 2 h at room temperature with the following primary antibodies: protein carbonyl primary antibody (Anti-DNP; # STA-308, Cell Biolabs) and SOD2 (#Ab13533, Abcam). After three 10 min washes in TBST, membranes were probed with secondary antibodies for protein carbonyl (HRP-conjugate; #STA-308, Cell Biolabs) and SOD2 (anti-rabbit IgG-HRP linked; #7074, Cell Signalling) in blocking solution for 2 h at room temperature and were finally extensively washed with TBST. The dilution of primary and secondary antibodies was optimised for each antibody. Chemiluminescence detection was carried out using ECL Clarity (Biorad), and images capture was done with ChemiDocMP. All the images were analysed using the Image Lab 4.0.1 software. Normalisation of protein signal intensities was carried out following the quantification of respective total protein levels on SF images, sample control, and internal standards.

### 2.10. NAD Measurement

Total NAD and NADH levels were measured in homogenised white gastrocnemius using a commercial kit (#K337-100, Biovision, USA) and following the manufacturer’s instructions. Briefly, the total pool of NAD (NADt = NAD^+^ and NADH) was extracted. For each of the extracted samples, half of the sample was heated to 60 °C for 30 min to decompose NAD^+^ while keeping NADH intact. Both NADt and NADH samples were mixed with NAD cycling enzyme, and absorbance was measured at 450 nm. NADt and NADH were quantified by comparing with NADH standard curve and normalising to mg of protein. Finally, the ratio of NAD^+^ (NADt—NADH) to NADH was calculated. All the assays were performed in triplicate.

### 2.11. Statistics

Data are reported as mean ± standard deviation (SD). Data normality was assessed using the D’Agostino-Pearson normality test. Data analysis was performed using the Student’s t-test and two-way ANOVA (group and time) for repeated measurements followed by Bonferroni’s *post hoc* test using Prism 8.4.1 (GraphPad Software, San Diego, CA, USA). A two-sided level of 5% for type I error was applied.

## 3. Results

### 3.1. Body Composition and Whole-Body Metabolism

Fifteen days of CF supplementation, in wild-type (WT) mice, did not modify body weight (*p* = 0.091), lean mass (*p* = 0.073) and fat mass assessed by EchoMRI (*p* = 0.89; Figure 1A). O_2_ consumption rates expressed per lean mass was significantly higher in the dark cycle (8 ± 8%, *p =* 0.042), whereas it remained similar in the light cycle (*p =* 0.392; Figure 1B.). There were no changes in locomotor activity (data not shown).CF mice showed a significant increase of whole-body energy expenditure by 15 ± 15% (*p =* 0.042; Figure 1C) and in food intake (0.08 ± 0.01 vs. 0.11 ± 0.01 g per lean body weight, respectively CF group and WT, *p =* 0.013, Figure 1D).

CF supplementation significantly modified whole-body metabolism: a main effect of higher RER in the CF group was observed over 24 h (*p* = 0.023, Figure 2A) and mean RER over 12 h was higher in the CF group in light and dark cycles (+5 ± 3%, *p* = 0.035 and +9 ± 4%, respectively, *p =* 0.026). Moreover, CF supplementation promoted carbohydrate (CHO) use (51 ± 13% of energy consumption depended on CHO oxidation in the CF group vs. 37 ± 12% in the control group during the light cycle, *p =* 0.108 and 71 ± 21% in CF group vs. 47 ± 24% in the control group during the dark cycle, *p =* 0.004; Figure 2B.). However, when expressed in kJ, there was no difference in CHO oxidation nor fatty acid oxidation (CHO oxidation: 3.7 ± 1.4 kcal·d^−1^ in control vs. 4.4 ± 1.7 kcal·lean body mass^−1^·d^−1^ in CF, *p =* 0.33 and fatty acid oxidation: 2.3 ± 1.0 kcal·d^−1^ in control vs. 2.6 ± 0.87 kcal·lean body mass^−1^·d^−1^ in CF, *p =* 0.608). Analysis of PRCF of RER revealed an increase of EC50 values in the CF group (*p =* 0,009; Figure 2D) meaning that CF induce a shift toward a greater use of carbohydrates. A decrease of the Hill slope was observed in the CF group (H values: 25.16 ± 4.44 in controls vs. 12.79 ± 3.96 in CF group, *p* < 0.001). Moreover, the RER amplitude over 24 h was increased by 31 ± 31% in the CF group (*p =* 0.035). CF supplementation improved glucose tolerance following oral glucose administration from unchanged baseline blood glucose levels after an overnight fast (Figure 2E, *p =* 0.003). The area under the capillary blood glucose curve was lower in the CF group (1018 ± 135 vs. 1181 ± 170 mM·120 min, *p =* 0.005; Figure 2F).

### 3.2. Mitochondrial Bioenergetics

We tested the effect of CF supplementation on the energetics of mitochondria in the white gastrocnemius and soleus muscle in WT mice. A higher mitochondrial respiration using complex IV substrates in the white gastrocnemius and soleus muscles was observed (*p =* 0.004 and *p =* 0.028, respectively; Figure 3A,B). No modification of the CII/CI and CIV/CI respiration ratios were observed (*p =* 0.899 and *p =* 0.701 for the white gastrocnemius and *p =* 0.693 and *p =* 0.912 for the soleus muscles, respectively, data not shown). Moreover, complex I, II, IV, and citrate synthase activities were significantly increased in the white gastrocnemius muscle (respectively: +31 ± 27%, *p =* 0.004; +28 ± 38%, *p =* 0.027; +35 ± 38%, *p =* 0.009; +14 ± 13%, *p =* 0.040; Figure 3C). The ability of mitochondria to oxidise palmitoyl-carnitine tends to increase in the white gastrocnemius (+24 ± 12%, *p =* 0.096; Figure 3D), whereas no difference was observed in the soleus (*p =* 0.363). The relative rates of palmitoyl-carnitine-stimulated respiration were unchanged when expressed in function of complex I respiration (62 ± 11 in the control group vs. 64 ± 20 in the CF group in the white gastrocnemius and 70 ± 18 vs. 72 ± 16 in the soleus). Next, we explored the sensitivity to Ca^2+^-induced PTP opening in the white gastrocnemius. As shown in Figure 3E, neither the time to PTP opening nor the Ca^2+^ retention capacity was modified following CF supplementation (*p =* 0.608 and *p =* 0.943, respectively).

mRNA assessment of genes involved in mitochondrial biogenesis showed a 70% upregulation of NRF1 mRNA, whereas no difference was reported for PGC1α, Tfam, CS, ND1, ND2, SDHa and Cox2 (Table 3). We explored, on a few samples, the effect of CF supplementation on mitochondrial supercomplexes (Appendix A). The densitometric analysis suggests that CF supplementation increased overall supercomplexes content without qualitative adaptation or rearrangements. Indeed, respirasome density was improved by 58% using complex I and complex II probes and by 93% using complex III probe. Similarly, complex I and complex III content embedded in supercomplexes were higher in the CF group.

CF supplementation lowers mitochondrial ROS emission. While H_2_O_2_ emission remained unchanged both in the white gastrocnemius and soleus when expressed per dry weight (Figure 4A,B, respectively), the rates were decreased when expressed per mitochondrial content (Figure 4C). A main positive effect of CF supplementation was observed in various conditions in the white gastrocnemius (*p* < 0.048; Figure 4C). The mitochondrial H_2_O_2_ emission was also significantly reduced when expressed per oxidative phosphorylation capacity assessed by state 4 respiration using complex I+II substrates by 26 ± 13% in the white gastrocnemius (*p =* 0.044) and by 29 ± 12% in the soleus (*p =* 0.024; Figure 4D).

### 3.3. Oxidative Stress Markers in Skeletal Muscle

The cellular protein oxidative stress level in WT mice was unaffected by CF supplementation as the cellular content of SOD2 was similar in the experimental and control groups in the white gastrocnemius and soleus (Figure 4E). Moreover, no difference was observed in protein carbonylation in the white gastrocnemius and soleus (Figure 4F). Additionally, no difference in catalase and MnSod mRNA expression was observed after CF supplementation (Table 3).

### 3.4. NAD Metabolism

The total pool of pyridine nucleotides increased following CF supplementation by 36 ± 33% (*p =* 0.012). Specifically, the NAD^+^ and NADH contents increased in the white gastrocnemius (respectively by 69 ± 60%, *p =* 0.026 and 29 ± 42%, *p =* 0.017; Figure 5A,B) and the NAD^+^/NADH ratio tended to increase in the CF group (*p =* 0.084). No significant effect was observed for Sirt3 mRNA (*p =* 0.096), NMNAT mRNA (*p =* 0.094), and Sirt1 mRNA(*p* = 0.869; Table 3).

### 3.5. Whole-Body and Cellular Metabolism Response Following CF Supplementation in Sirt3^-/-^ Mice

We examined the effect of CF supplementation in Sirt3^-/-^ mice. Body composition as weight, lean mass, and fat mass were not different between the CF and control groups (*p =* 0.52, *p =* 0.66, *p =* 0.45, respectively; Figure 6A). Similarly, the daily energy expenditure was unaffected by CF supplementation (0.38 ± 0.01 Kcal·lean body mass^−1^·day^−1^ in the controls vs. 0.39 ± 0.01 Kcal·lean body mass^−1^·day^−1^ in CF, *p =* 0.502; Figure 6B). The RER was similar between both groups in the light and dark cycles (*p =* 0.301; Figure 6C), and substrate oxidation was not affected by CF supplementation (44 ± 17% of energy consumption depended on CHO oxidation in the control group vs. 50 ± 11% in the CF group during the light phase, *p =* 0.209 and 65 ± 19% in the control group vs. 77 ± 21% in the CF group during the dark phase, *p =* 0.395; Figure 6D). The metabolic flexibility assessed by RER amplitude over 24 h remained similar between the two groups (0.19 ± 0.05 in control vs. 0.20 ± 0.03 in CF, *p =* 0.604), whereas the PRCF slope was decreased in the CF group (H values: 17.77 ± 4.38 in control vs. 12.79 ± 3.93 in CF, *p =* 0.039).

Next, we compared the effect of CF supplementation on mitochondrial enzyme activities. CF supplementation failed to increase the enzyme activity of complex I, II, IV, and CS in Sirt3^-/-^ mice as observed previously in WT (respectively: *p =* 0.623, *p =* 0.629, *p =* 0.283, and *p =* 0.791, Figure 6E). A greater enzyme activity of complex I and IV was observed in WT compared to Sirt3^-/-^ mice after CF supplementation (respectively: *p* = 0.027 and *p* = 0.045)

## 4. Discussion

In this study, we provide an integrative analysis of the effect of CF ingestion. We observed that chronic CF ingestion improved mitochondrial respiration and reduced H_2_O_2_ production, assessed in their myofibre environment, whereas the susceptibility of PTP opening was unchanged. CF ingestion enhanced NAD metabolism, supporting the involvement of sirtuin pathways on the observed mitochondrial adaptations. We also observed whole-body metabolism modifications towards a greater ability to use carbohydrates as a main substrate suggesting that mitochondrial adaptations likely contributed to the observed whole-body metabolism adaptation.

### 4.1. Metabolism

Dietary administration of cocoa flavanols has been identified as an effective strategy for alleviating glucose intolerance, allowing opportunities to ameliorate metabolic diseases [34]. Our results are in accordance with previous studies that reported an increased glucose tolerance following flavanols intake [35,36]. The mechanisms proposed for this effect involve an increase in the translocation of GLUT4 towards the cell membrane, an increase in the phosphorylation of AMPK and the up-regulation of UCP-2 gene expression in the skeletal muscle [8,37]. Moreover, CF ingestion was associated with insulin sensitivity improvement in humans [8]. After 15 days of dark chocolate consumption, a decrease of the homeostasis model assessment of insulin resistance index was observed in healthy subjects concomitantly with an increase of the quantitative insulin sensitivity check index and the insulin sensitivity index [8]. We also observed a concomitant increase in glucose tolerance and metabolic flexibility in WT mice, which refers to the the organism’s capacity to adapt fuel oxidation to fuel availability [38]. Interestingly, while impaired mitochondrial oxidative capacity and the related substrate oxidation was associated with insulin resistance [39], endurance exercise training increased mitochondrial content, insulin sensitivity, and metabolic flexibility [38]. Therefore, the present results are specific to CF supplementation and are not influenced by spontaneous activity.

The positive effect of CF on metabolic flexibility, as well as the increase of mitochondrial complexes activity, were blunted in Sirt3^-/-^ mice, indicating that the positive effects of CF occur at least partially through Sirt3. These results support previous findings and a pivotal role of Sirt3 on metabolic flexibility regulation [24]. Our results suggest that an increase in mitochondrial function may enhance metabolic flexibility. However, besides mitochondrial function, the basal respiratory quotient, glucose disposal rate, adipose tissue lipid storage capacity, and plasma free fatty acid concentration were also identified as determinant factors of metabolic flexibility and should be considered [40]. Future studies should assess the cellular determinants of metabolic flexibility more deeply to deepen our understanding of the underlying causes of enhancement of metabolic flexibility following CF supplementation.

The whole-body metabolism assessment revealed an increase in the relative CHO oxidation following CF supplementation. This result contrasts with a previous study that reported an increase of lipolysis following an acute flavanols ingestion or a two-week supplementation with flavan-3-ols in mice [36]. Besides the extract composition differences, these contradictory findings may result from the pleiotropic effects of NO. It is well accepted that CF ingestion stimulates the production or the bioavailability of NO metabolism [7]. On one side, NO stimulates glucose transport and uptake in skeletal muscle. Stimulation of NO production in cell line-derived myotubes by insulin or hydrogen peroxide resulted in an increase of GLUT4 translocation, which was reduced with nNOS inhibition [41]. Moreover, NO can increase glucose transport in an insulin-independent pathway leading to an increase in the levels of cyclic GMP and AMPK [42], which promotes glucose oxidation in rat skeletal muscle independent of fatty acid availability and oxidation [43]. On the other side, NO promotes fatty acid oxidation by reducing the level of malonyl-CoA via inhibition of acetyl-CoA carboxylase and activation of malonyl-CoA decarboxylase in skeletal muscle [44]. While we observed a shift towards a lower relative proportion to use fatty acid as substrate with CF, there was no difference regarding the absolute values expressed in kJ. This suggests that the increase in energy expenditure observed with CF supplementation relies on an increase in glucose oxidation. Therefore, these results support the idea that CF consumption increases insulin sensitivity and promotes glucose oxidation without a concomitant effect on fatty acid oxidation capacity. These results raise the question of the relevance to use cocoa flavanols in the context of high fat diet. Future studies should evaluate a possible harmful effect of an association of a CF supplementation and a high-fat diet.

### 4.2. Mitochondrial Function

Dietary administration of cocoa flavanols has been shown to improve mitochondrial function, including (i) mitochondrial respiration and electron transport chain enzyme activities; (ii) mitochondrial H_2_O_2_ release, which reflects mitochondrial reactive oxygen species (ROS) production; and (iii) mitochondrial calcium retention capacity which reflects the susceptibility to mitochondrial permeability transition pore (mPTP) opening (for review see [11]. Our results are in accordance with previous studies; nonetheless, our study was the first to evaluate the effect of CF supplementation on mitochondrial function assessed in situ in permeabilised fibre and in different muscle types. Unlike the previous studies on isolated mitochondria, the assessment of mitochondrial function in permeabilised fibre preserves mitochondrial morphology, functional interactions with other intracellular components [1] and avoids mitochondrial structure disruption during the mitochondrial isolation process [32]. Therefore, it is important to study mitochondrial function in tissue preparations where mitochondrial structure is preserved, and all of the mitochondrial pool is represented to better evaluate the response to a treatment.

Previous studies performed in cell culture or in vivo in mice have shown that supplementation with EPI, which is considered the main bioactive molecule in CF, increases maximal ADP-stimulated respiration (e.g., state 3 respiration) when mitochondria are energised with a combination of energy substrates feeding various sites along the respiratory chain [45,46,47,48,49]. Consistent with these results, we observed a main effect of CF supplementation on mitochondrial respiration in oxidative and glycolytic muscles. To determine whether this increase is related to a mitochondrial mass increase or the remodelling of mitochondrial respiratory complexes, we assessed mitochondrial supercomplexes from the quadriceps muscle. These supramolecular structures are thought to decrease ROS production, stabilise or assist in the assembly of individual complexes, regulate respiratory chain activity, and prevent protein aggregation in the protein-rich inner mitochondrial membrane [50,51,52]. Despite a limited number of measures, the visual analysis of the results (Appendix A) suggests that CF supplementation induces an increase of mitochondrial mass, as reflected by an increase in all supercomplex species, rather than specific respiratory chain complex reorganisations. Collectively, these results suggest that CF promotes a mitochondrial mass increase without qualitative adaptation of respiratory chain complex arrangements.

During mitochondrial respiration, variable amounts of superoxide are formed and can be metabolised to form other types of ROS [53]. Although excessive ROS production from mitochondria is involved in a broad spectrum of pathologies, low physiological concentrations of ROS can have beneficial effects, such as protecting against infectious agents and participating in cellular signalling [2,54]. Previous studies performed in vivo in rodents and humans and cell culture models reported beneficial CF supplementation effects on ROS production and various antioxidant systems [11]. For instance, oral gavage with EPI during 15 days increases SOD2 and catalase activity in mice quadriceps muscles [17]. While mitochondrial ROS production has not been widely assessed, we observed that ROS production levels were similar when values were expressed per muscle weight, whereas the production was lowered when expressed per mitochondrial mass, suggesting that qualitative adaptations occur to maintain the physiological concentrations of ROS. Moreover, we did not observe any effect of CF on SOD2 nor ROS-induced damages assessed by protein carbonylation. While CF enhances antioxidant protein pools, including widely reported SOD2 and catalase levels [16,17,55], this adaptation was not always associated with a decrease in ROS-induced damages [56,57,58,59]. These data support the theory that low levels of ROS production are required for normal cell homeostasis [2]. Collectively, these results suggest that, concomitantly to an increase of mitochondrial mass, CF promotes qualitative mitochondrial adaptations to enhance its antioxidant capacity and maintain low levels of ROS production.

### 4.3. Mitochondrial Biogenesis

Mitochondrial mass is the result of the interplay between mitochondrial biogenesis and mitophagy. Our study focused on mitochondrial biogenesis activated by polyphenols consumption [60]. Mitochondrial biogenesis is the cellular process that improve mitochondrial mass. This complex and highly regulated process involve several transcription factors, nuclear hormone receptors, and transcription coactivators that act collectively to coordinate changes in the expression of nuclear and mitochondrial DNA encoded genes [61]. Previous studies have attempted to identify which pathway(s) may underlie CF supplementation-induced mitochondrial biogenesis. Stimulation of NO-dependent signalling has emerged as a central pathway involved in mitochondrial biogenesis. CF supplementation enhances the generation of NO by inhibiting the arginine degrading enzyme arginase, which increases the availability of L-arginine for NO biosynthesis [62,63]. Moreover, using an eNOS inhibitor, the stimulatory effect of epicatechin on mitochondrial biogenesis was partially blunted [17]. These results suggest the involvement of other signalling pathways to stimulate mitochondrial biogenesis.

Recently, Aragones et al. (2016) reported that dietary proanthocyanidins, members of the polyphenols family, boost hepatic NAD^+^ metabolism by enhancing the *de novo* NAD^+^ biosynthesis pathway and Sirt1 activity in a dose-dependent manner in rats [12]. Furthermore, the liver exports NAD^+^ precursors to other organs, such as skeletal muscle, which have a lower capacity to synthetise NAD, raising the possibility of extending the effects observed in the liver to the whole organism [64]. While some studies reported an effect of CF supplementation on Sirt1 protein content, activity, or mRNA [12,15,16,17,65,66], the involvement of NAD metabolism in mitochondrial biogenesis following CF supplementation using loss or gain of function has never been tested. While a muscle-specific knock-out would have been preferable over the whole-body Sirt3 KO mice used in our study, our results suggest that CF improves NAD metabolism, and Sirt3 is involved in this supplementation-induced mitochondrial electron transport chain activity improvement. The positive effects observed on mitochondrial complexes activity following CF supplementation were not observed using Sirt3^-/-^ mice and a greater activity of complex I and IV activities were observed after CF supplementation only in WT. These observations extend previous works showing that Sirt3 plays a pivotal role in mitochondrial function [67]. Collectively, these findings suggest that sirtuins, through the modulation of NAD metabolism, are involved in mitochondrial mass improvement observed following CF supplementation.

### 4.4. Limits

Cocoa beans are composed by a mixture of monomeric, oligomeric, and polymeric flavanols. Among the flavanols, EPI appears to be the most abundant polyphenolic monomer found in cacao products, representing up to 35% of the polyphenol content [68]. EPI was established as the main bioactive molecule underlying benefits associated with cocoa and chocolate supplementation [47,69]. However, EPI metabolites (EPIm) may be involved in the EPI-induced biological effects observed in in vivo studies. EPIm can reach higher plasma concentration than EPI [70]. Moreover, major differences in EPIm were observed across species. Indeed, 80% of EPIm present in humans were not detected in rats, while the similarity between mice and humans was found to be slightly more favourable as the two major human metabolites were detected [70]. We used enriched CF extract containing up to five times more EPI than unfermented cocoa beans [71]. Moreover, EPI content was reduced by 10–20% by fermentation and decrease in a temperature-dependent manner during roasting [71,72]. Collectively, these data suggest that attention should only be paid on the EPI content when assessing the benefits of EPI supplementation on human health, and the present results should be considered with caution before extrapolating the observed underlying mechanisms in humans.

## 5. Conclusions

Daily CF supplementation for 15 days leads to mitochondrial function improvements in oxidative and glycolytic muscles. The present results suggest that the increased mitochondrial respiration results in an increase in mitochondrial mass while the cellular level of ROS production was maintained, suggesting qualitative adaptations within the mitochondria. Concomitant to the cellular effect on NO metabolism previously described [73], CF modulates NAD metabolism, which stimulates sirtuins activity. Sirt3 plays a central role in mitochondrial adaptations and metabolic flexibility following CF supplementation. As sirtuins are recent targets in various disease pathogenesis, such as diabetes or asthma [74,75], further studies will explore the potential interest of using CF as a natural activator of sirtuins.

## Figures and Tables

**Figure 1 nutrients-13-03466-f001:**
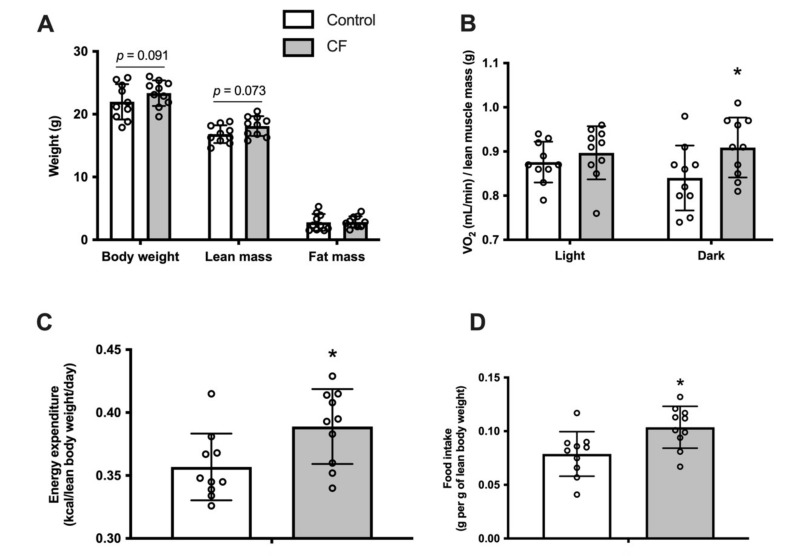
Fifteen days of CF supplementation influences body composition and increases energy expenditure in 129S1/SvlmJ mice. (**A**) Body weight, lean mass, and fat mass. (**B**) Measurement of whole-body oxygen consumption during light and dark phases. (**C**) Daily energy expenditure. (**D**) Daily food intake. *n* = 10, means ± SD. * *p* < 0.05 *vs*. the control group.

**Figure 2 nutrients-13-03466-f002:**
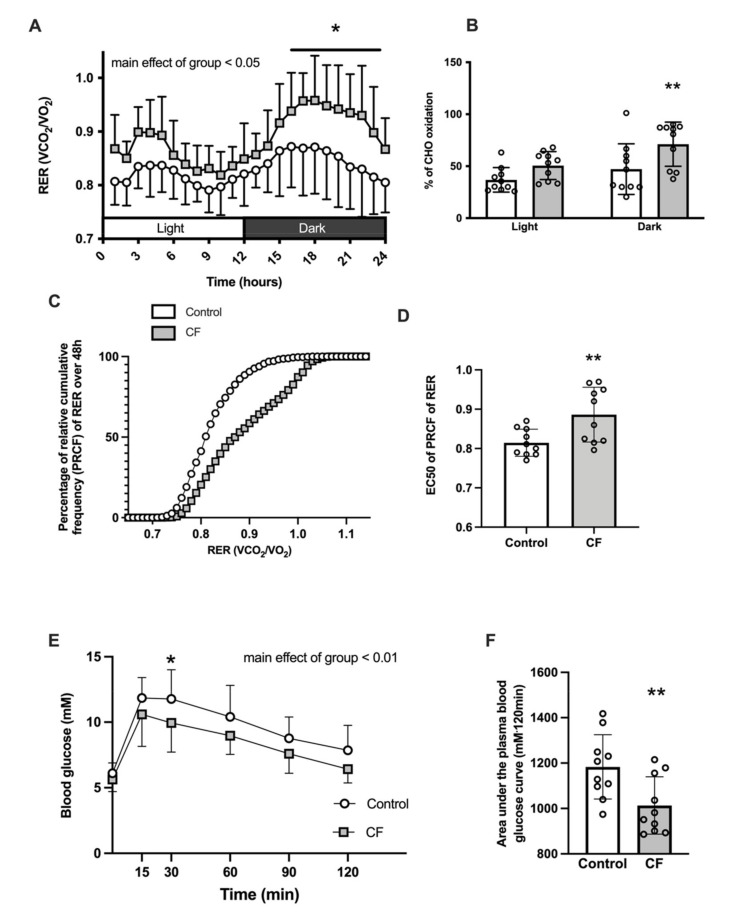
Fifteen days of CF supplementation modified whole-body metabolism in 129S1/SvlmJ mice. (**A**) Respiratory exchange ratio (RER) over the 24 h cycle. (**B**) Carbohydrate and fat oxidation in light and dark cycles. (**C**) Percent relative cumulative frequency (PCRF) of RER. (**D**) Fiftieth percentile values (EC50) of PCRF of RER (**E**) OGTT after an overnight fasting. (**F**) Area under the curve of the OGTT. *n* = 10–14, means ± SD. * *p* < 0.05 vs. the control group, ** *p* < 0.01 vs. the control group.

**Figure 3 nutrients-13-03466-f003:**
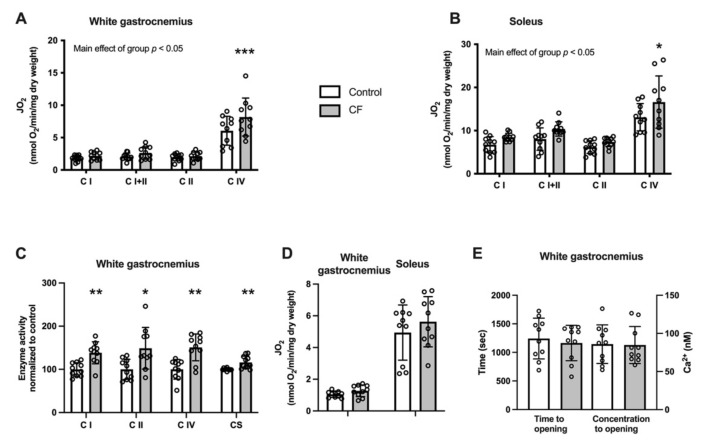
Fifteen days of supplementation improved mitochondrial respiration and enzyme activities but did not modify mitochondrial sensitivity to Ca^2+^ in 129S1/SvlmJ mice. (**A**) Respiration rates (JO_2_) expressed per unit of dry weight from the white gastrocnemius muscle. (**B**) Enzyme activities of respiratory chain complexes and citrate synthase in the white gastrocnemius muscle. (**C**) Respiration rates (JO_2_) using palmitoyl-carnitine as substrate in the white gastrocnemius and soleus muscles. (**D**) Effects on the time to PTP opening and the Ca^2+^ retention capacity in the white gastrocnemius muscle. (**E**) Respiration rates (JO_2_) expressed per unit of dry weight from the soleus muscle. *n* = 10, means ± SD. * *p* < 0.05 vs. the control group, ** *p* < 0.01 vs. the control group, *** *p* > 0.001 vs. the control group.

**Figure 4 nutrients-13-03466-f004:**
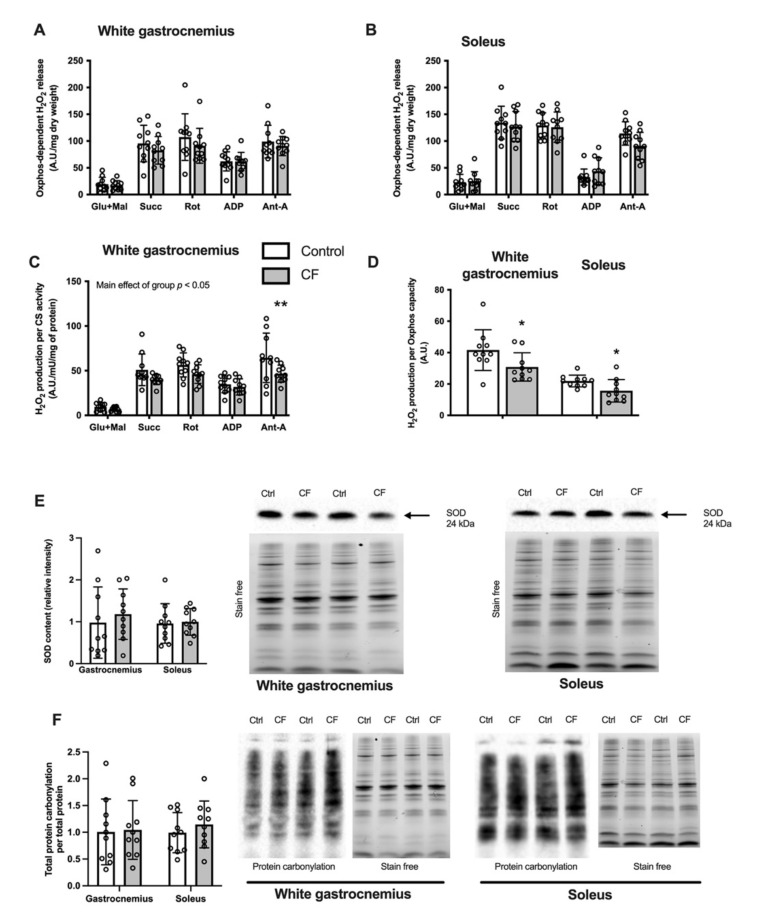
Fifteen days of supplementation modified ROS metabolism in 129S1/SvlmJ mice. (**A**) Rates of H_2_O_2_ production expressed per unit of dry weight from the white gastrocnemius muscle. (**B**) Rates of H_2_O_2_ production expressed per unit of the marker enzyme citrate synthase (CS) from the white gastrocnemius muscle. (**C**) Rates of H_2_O_2_ production expressed per unit of dry weight from soleus muscle. (**D**) Maximal rate of H_2_O_2_ production expressed per Oxphos capacity from the white gastrocnemius and soleus muscle. (**E**) SOD2 content in the white gastrocnemius and soleus muscles, and representative examples of SOD2 and stain free total protein. (**F**) Total protein carbonylation content per total protein content and representative examples of protein carbonylation (detected as ubiquitin-conjugated proteins) in the white gastrocnemius and soleus muscles. *n* = 8–10, means ± SD. * *p* < 0.05 vs. the control group, ** *p* < 0.01 vs. the control group.

**Figure 5 nutrients-13-03466-f005:**
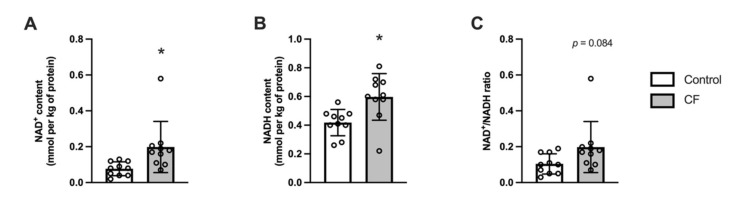
Fifteen days of supplementation modified NAD^+^ metabolism in the white gastrocnemius in 129S1/SvlmJ mice. (**A**) NAD^+^ content. (**B**) NADH content. (**C**) NAD^+^/NADH ratio. *n* = 10, means ± SD. * *p* < 0.05 vs. the control group.

**Figure 6 nutrients-13-03466-f006:**
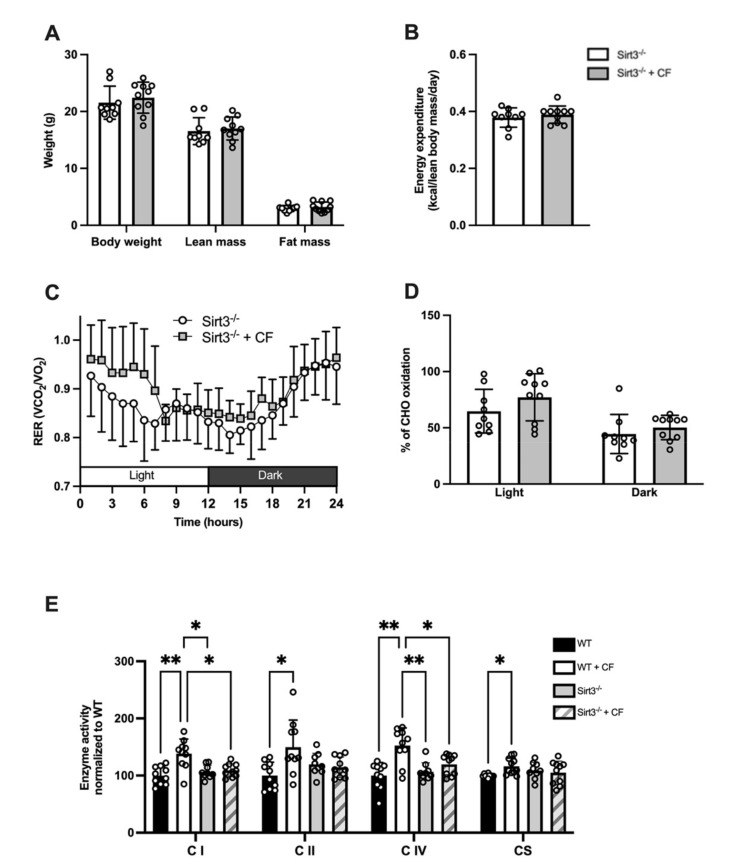
Impact of 15 days of CF supplementation on whole-body and mitochondrial metabolism of Sirt3^-/-^ mice. (**A**) Body weight, lean mass, and fat mass. (**B**) Daily energy expenditure. (**C**) Respiratory exchange ratio (RER) over a 24 h cycle. (**D**) Carbohydrate and fat oxidation in light and dark cycles. (**E**) Enzyme activities of respiratory chain complexes and citrate synthase in the white gastrocnemius muscle. *n* = 10 in the control group and *n* = 9 in the Sirt3^-/-^ group, means ± SD, * *p* < 0.05, ** *p* < 0.01.

**Table 1 nutrients-13-03466-t001:** Composition of the cocoa powder.

Compound	Content
Total flavanols (%)	31.62
(-)-epicatechin (%)	5.93
(-)-catechin (%)	1.21
Théobromin (%)	7.12
Caffeine (%)	0.92

**Table 2 nutrients-13-03466-t002:** Primer sequences used in real-time PCR analysis.

Target Gene	Official Full Name	GenBank Accession Number	Forward Primer (5′-3′) Reverse Primer (3′-5′)
PGC1α	Peroxisome proliferative activated receptor, gamma, coactivator 1 alpha	NM_008904.2	AAACTTGCTAGCGGTCCTCA TGGCTGGTGCCAGTAAGAG
NRF1	Mus musculus nuclear respiratory factor 1	NM_001164226	GCACCTTTGGAGAATGTGGT GGGTCATTTTGTCCACAGAGA
TFAM	Mus musculus transcription factor A, mitochondrial	NM_009360	CCTTCGATTTTCCACAGAACA GCTCACAGCTTCTTTGTATGCTT
CS	Citrate synthase	NM_026444.4	GGAGCCAAGAACTCATCCTG TCTGGCCTGCTCCTTAGGTA
ND1	NADH dehydrogenase subunit 1	NC_005089.1	ACACTTATTACAACCCAAGAACACAT TCATATTATGGCTATGGGTCAGG
ND2	NADH dehydrogenase subunit 2	NC_005089.1	CCATCAACTCAATCTCACTTCTATG GAATCCTGTTAGTGGTGGAAGG
SDHa	Succinate dehydrogenase complex, subunit A	NM_023281.1	GGAACACTCCAAAAACAGACCT CCACCACTGGGTATTGAGTAGAA
Cox2	Cytochrome c oxidase subunit II	NC_005089.1	CATCTGAAGACGTCCTCCACTCAT TGCTTGATTTAGTCGGCCTGGGAT
Cat	Catalase	NM_009804.2	TGAGAAGCCTAAGAACGCAATTC CCCTTCGCAGCCATGTG
MnSOD	Superoxide dismutase [Mn]	NM_013671.3	TTAACGCGCAGATCATGCA GGTGGCGTTGAGATTGTTCA
Sirt1	Sirtuin 1	NM_019812	AAAGGAATTGGTTCATTTATCAGAG TTGTGGTTTTTCTTCCACACA
Sirt3	Sirtuin 3	NM_022433.2	AGGTGGAGGAAGCAGTGAGA GCTTGGGGTTGTGAAAGAAA
NMNAT1	Nicotinamide nucleotide adenyltransferase 1	NM_133435.2	TGTGCCCAAGGTGAAATTGCT CCACGATTTGCGTGATGTCC
GAPDH	glyceraldehyde-3-phosphate dehydrogenase	NM_008084.3	ACTCCACTCACGGCAAATTC GTTAGTGGGGTCTCGCTCCT

**Table 3 nutrients-13-03466-t003:** mRNA levels of genes involved in mitochondrial biogenesis, anti-oxidant defences and NAD metabolism. The mRNA levels were determined by real-time quantitative RT-PCR and expressed relative to the value of GAPDH, an internal control. Data are presented as the mean ± SD, *n* = 8.

Gene	Control Group	Group	*p*
PGC1α	1.00 ± 0.42	1.21 ± 1.28	0.66
NRF1	1.00 ± 0.42	1.70 ± 0.72	0.04
TFAM	1.00 ± 0.69	1.27 ± 0.52	0.45
CS	1.00 ± 0.20	1.34 ± 0.49	0.08
ND1	1.00 ± 1.02	1.23 ± 0.79	0.64
ND2	1.00 ± 1.16	1.32 ± 0.72	0.53
SDHa	1.00 ± 0.37	1.06 ± 0.53	0.80
Cox2	1.00 ± 0.92	1.37 ± 0.86	0.44
Catalase	1.00 ± 0.91	0.56 ± 0.32	0.22
MnSOD	1.00 ± 0.84	0.74 ± 0.32	0.45
Sirt1	1.00 ± 1.04	0.92 ± 0.76	0.86
Sirt3	1.00 ± 0.46	1.93 ± 1.36	0.10
NMNAT	1.00 ± 0.38	1.34 ± 0.38	0.10

## Data Availability

The data that support the findings of this study are available from the corresponding author upon reasonable request.

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
