# Peer review of "Dietary Cocoa Flavanols Enhance Mitochondrial Function in Skeletal Muscle and Modify Whole-Body Metabolism in Healthy Mice"

_nutrients, 2021, doi:10.3390/nu13103466_

Round 1

Reviewer 1 Report

In this study, authors provide an integrative analysis of the effect of 15 days of CF supplementation in wild type and Sirt3-/- 17 mice. Whole body metabolism was modified following a 15-day CF supplementation towards a greater ability to use carbohydrates as a main substrate. Authors observed that chronic CF ingestion improved mitochondrial respiration and reduced H2O2 production, assessed in their myofiber environment. Authors observed that CF supplementation boosted the NAD metabolism that stimulates sirtuins metabolism and improved mitochondrial function, which likely contributed to the observed whole-body metabolism adaptation, with a greater ability to use carbohydrates, at least partially through Sirt3. Generaly artice is well writen and read very good. Most of the presented result are presented clearly bu some minor corrections shuld be made. Introduction: Why authors chosen 15 day suplementation, whether the time of supplementation was supported by previous studies, or was chosen or was selected randomly? Furthermore authors write that Sirt3 plays an important role in the regulation of whole body metabolism, but in the contex of this publication it is urgent to write something more about the mechanism of this regulation. Methods Animals and diet Authors chosen one week of habituation in the animal facility for the animals - according to the medical models of pets behavior one week period may be in some situtation to short and it may reflects in the results. How composition of the cocoa powder was analysed? Results: Authors write about mRNA levels of genes involved in mitochondrial biogenesis, anti-oxidant defences and NAD metabolism. There is no explanation of the abbreviations of the names used in the table.

Author Response

The authors would like to thank the reviewers for taking the time to carefully evaluate our manuscript and the constructive feedback they have provided. We believe that this feedback has allowed us to greatly improve the clarity and quality of our manuscript. Kindly refer to the itemized point-by-point response.

Reviewer #1

Comment 1:

Introduction: Why authors chosen 15 day suplementation, whether the time of supplementation was supported by previous studies, or was chosen or was selected randomly?

Response:

The supplementation has been determined based on previous studies that explore the effects of (-)-epicatechin on mitochondria. In mice, treatment ranges from 10 to 15 days (Moreno-Ulloa et al., 2015; Nogueira et al., 2011; Panneerselvam et al., 2013; Ramirez-Sanchez et al., 2014; Ramírez-Sánchez et al., 2016; Watanabe et al., 2014). Therefore, we decided to test a 15-day treatment.

Moreover, the treatment was determined using the bioavailability of EPI and previous studies that explored effects of (-)-epicatechin on mitochondria. The ingestion of radiolabeled [14C]-epicatechin in humans showed that ~80% of ingested [14C]-epicatechin was absorbed (Ottaviani et al., 2016). The plasma radioactivity kinetic profile was biphasic, with peaks at 1 and 6 hours supporting the interest of the ingestion of two dose per day instead of one dose per day. Although supraphysiological concentrations (10-100µM) have been used, clear evidence of positive effects has been observed using (-)-epicatechin concentrations in the low molecular range (as low as 1µM), which is close to the concentrations encountered in vivo (Chidambaram et al., 2018). We are currently using enriched cocoa in human studies with positive effects. We therefore decided to use similar dose in mice. We used guidance for industry providing common conversion factors for deriving a human equivalent dose in mice (Alert, 2005).

Comment 2:

Furthermore authors write that Sirt3 plays an important role in the regulation of whole body metabolism, but in the contex of this publication it is urgent to write something more about the mechanism of this regulation.

Response:

We did not clearly identify what is the expectation. To our point of view, the following sentences (l. 68-77) describe the interest to explore Sirt3 pathway and describe the general mechanism of action (deacetylation): « Sirt3, which is highly expressed in tissues with high metabolic turnover and mitochondrial content, is of particular interest concerning its critical role in maintaining normal mitochondrial function through reversible protein lysine deacetylation (Ahn et al., 2008; Hirschey et al., 2010; Schwer et al., 2006). Moreover, Sirt3 plays an important role in the regulation of whole-body metabolism (Ahn et al., 2008). Sirt3 was shown to be altered in skeletal muscle of models of type 1 and type 2 diabetes and cardiovascular diseases (Jing et al., 2011; Wu et al., 2019). Furthermore, it was suggested that activation of Sirt3 might represent a promising therapeutic strategy for the improving mitochondrial function and metabolism (Jing et al., 2013; Wu et al., 2019). Therefore, determining whether Sirt3 underlies part of the beneficial effects of CF supplementation on whole-body metabolism and mitochondrial function is of interest. »

Comment 3:

Methods Animals and diet Authors chosen one week of habituation in the animal facility for the animals - according to the medical models of pets behavior one week period may be in some situation to short and it may reflects in the results.

Response:

The animals remained in the facility at least one week before the beginning of the experiments. Two animals were included every day (one control and one experimental animal) meaning that the animals involved in the study began the experiment between one week and 2.5 weeks after their arrival in the facility. We added an information in the methods section (l.93) to precise that the duration of habituation was at least one week. Moreover, the effects of the supplementation were measured after 15 days and we could assume that the interference between the beginning of the supplementation and the duration of habituation is limited.

Comment 4:

How composition of the cocoa powder was analysed?

Response:

The cocoa powder composition has been performed using high performance liquid chromatography by the manufacturer. The information was added in the manuscript, please read as follow: “The composition has been established by high performance liquid chromatography method and 100mg of extract are equivalent to an average of 475mg of dry cocoas seeds.”

Comment 5:

Results: Authors write about mRNA levels of genes involved in mitochondrial biogenesis, anti-oxidant defences and NAD metabolism. There is no explanation of the abbreviations of the names used in the table. 

Response:

We did not add information to explain the abbreviations as we provide the official full name of the genes abbreviated in the second column of the table.

References

Ahn, B.-H., Kim, H.-S., Song, S., Lee, I.H., Liu, J., Vassilopoulos, A., Deng, C.-X., Finkel, T., 2008. A role for the mitochondrial deacetylase Sirt3 in regulating energy homeostasis. Proc. Natl. Acad. Sci. U.S.A. 105, 14447–14452. https://doi.org/10.1073/pnas.0803790105

Alert, O., 2005. Guidance for Industry Estimating the Maximum Safe Starting Dose in Initial Clinical Trials for Therapeutics in Adult Healthy Volunteers.

Chidambaram, S.B., Bhat, A., Ray, B., Sugumar, M., Muthukumar, S.P., Manivasagam, T., Justin Thenmozhi, A., Essa, M.M., Guillemin, G.J., Sakharkar, M.K., 2018. Cocoa beans improve mitochondrial biogenesis via PPARγ/PGC1α dependent signalling pathway in MPP+ intoxicated human neuroblastoma cells (SH-SY5Y). Nutr Neurosci 1–10. https://doi.org/10.1080/1028415X.2018.1521088

Hirschey, M.D., Shimazu, T., Goetzman, E., Jing, E., Schwer, B., Lombard, D.B., Grueter, C.A., Harris, C., Biddinger, S., Ilkayeva, O.R., Stevens, R.D., Li, Y., Saha, A.K., Ruderman, N.B., Bain, J.R., Newgard, C.B., Farese, R.V., Alt, F.W., Kahn, C.R., Verdin, E., 2010. SIRT3 regulates mitochondrial fatty-acid oxidation by reversible enzyme deacetylation. Nature 464, 121–125. https://doi.org/10.1038/nature08778

Jing, E., Emanuelli, B., Hirschey, M.D., Boucher, J., Lee, K.Y., Lombard, D., Verdin, E.M., Kahn, C.R., 2011. Sirtuin-3 (Sirt3) regulates skeletal muscle metabolism and insulin signaling via altered mitochondrial oxidation and reactive oxygen species production. Proc. Natl. Acad. Sci. U.S.A. 108, 14608–14613. https://doi.org/10.1073/pnas.1111308108

Jing, E., O’Neill, B.T., Rardin, M.J., Kleinridders, A., Ilkeyeva, O.R., Ussar, S., Bain, J.R., Lee, K.Y., Verdin, E.M., Newgard, C.B., Gibson, B.W., Kahn, C.R., 2013. Sirt3 regulates metabolic flexibility of skeletal muscle through reversible enzymatic deacetylation. Diabetes 62, 3404–3417. https://doi.org/10.2337/db12-1650

Moreno-Ulloa, A., Nogueira, L., Rodriguez, A., Barboza, J., Hogan, M.C., Ceballos, G., Villarreal, F., Ramirez-Sanchez, I., 2015. Recovery of Indicators of Mitochondrial Biogenesis, Oxidative Stress, and Aging With (-)-Epicatechin in Senile Mice. J. Gerontol. A Biol. Sci. Med. Sci. 70, 1370–1378. https://doi.org/10.1093/gerona/glu131

Nogueira, L., Ramirez-Sanchez, I., Perkins, G.A., Murphy, A., Taub, P.R., Ceballos, G., Villarreal, F.J., Hogan, M.C., Malek, M.H., 2011. (-)-Epicatechin enhances fatigue resistance and oxidative capacity in mouse muscle. J. Physiol. (Lond.) 589, 4615–4631. https://doi.org/10.1113/jphysiol.2011.209924

Ottaviani, J.I., Borges, G., Momma, T.Y., Spencer, J.P.E., Keen, C.L., Crozier, A., Schroeter, H., 2016. The metabolome of [2-(14)C](-)-epicatechin in humans: implications for the assessment of efficacy, safety, and mechanisms of action of polyphenolic bioactives. Sci Rep 6, 29034. https://doi.org/10.1038/srep29034

Panneerselvam, M., Ali, S.S., Finley, J.C., Kellerhals, S.E., Migita, M.Y., Head, B.P., Patel, P.M., Roth, D.M., Patel, H.H., 2013. Epicatechin regulation of mitochondrial structure and function is opioid receptor dependent. Mol Nutr Food Res 57, 1007–1014. https://doi.org/10.1002/mnfr.201300026

Ramirez-Sanchez, I., De los Santos, S., Gonzalez-Basurto, S., Canto, P., Mendoza-Lorenzo, P., Palma-Flores, C., Ceballos-Reyes, G., Villarreal, F., Zentella-Dehesa, A., Coral-Vazquez, R., 2014. (-)-Epicatechin improves mitochondrial-related protein levels and ameliorates oxidative stress in dystrophic δ-sarcoglycan null mouse striated muscle. FEBS J. 281, 5567–5580. https://doi.org/10.1111/febs.13098

Ramírez-Sánchez, I., Rodríguez, A., Moreno-Ulloa, A., Ceballos, G., Villarreal, F., 2016. (-)-Epicatechin-induced recovery of mitochondria from simulated diabetes: Potential role of endothelial nitric oxide synthase. Diab Vasc Dis Res 13, 201–210. https://doi.org/10.1177/1479164115620982

Schwer, B., Bunkenborg, J., Verdin, R.O., Andersen, J.S., Verdin, E., 2006. Reversible lysine acetylation controls the activity of the mitochondrial enzyme acetyl-CoA synthetase 2. Proc. Natl. Acad. Sci. U.S.A. 103, 10224–10229. https://doi.org/10.1073/pnas.0603968103

Watanabe, N., Inagawa, K., Shibata, M., Osakabe, N., 2014. Flavan-3-ol fraction from cocoa powder promotes mitochondrial biogenesis in skeletal muscle in mice. Lipids Health Dis 13, 64. https://doi.org/10.1186/1476-511X-13-64

Wu, J., Zeng, Z., Zhang, W., Deng, Z., Wan, Y., Zhang, Y., An, S., Huang, Q., Chen, Z., 2019. Emerging role of SIRT3 in mitochondrial dysfunction and cardiovascular diseases. Free Radic. Res. 53, 139–149. https://doi.org/10.1080/10715762.2018.1549732

Reviewer 2 Report

The manuscript by Daussin F.N, et al “Dietary cocoa flavanols enhance mitochondrial function and 2 improve glucose tolerance in healthy mice” describes the cocoa flavanols supplementation to the healthy mouse for 15 days orally, which leads to improved glucose tolerance and mitochondrial function.

Major issues found in this study:

  1. The authors have Introduced epicatechin as a major component of cocoa powder, as per table 1. Theobromine is a major component (7.12%). They could have mention cocoa powder extract information ie., components in extraction, how you perform the extraction from cocoa powder. And how you decided the dose, 302.1mg/kg/B.W?
  2. 15-days of supplementation on the healthy mouse is not a good choice of model to see the effect. Why you have not studied the effect on the high-fat diet-fed mouse, where glucose tolerance is a major problem. Also, long-term treatment for 30-60 days may give significant changes.
  3. Article title to results is not matching, the title is very general. But you have only studied gastrocnemius and soleus muscles. The major metabolic organs are not involved in this study like liver, pancreas.
  4. All the enzyme activity you studied is on frozen tissues samples, It is a total tissue protein not mitochondria alone.
  5. 1: the effect of CF supplementation is very minimal with a broad range of differences between mice. There is no pathological (either diabetes or high-fat diet-fed mice) control
  6. 2: Panel A. RER below 0.7 is calories from fat. Above 0.7 is carbohydrate. Both control and CF supplement groups using carbs as their fuel source. According to this result mouse supplemented with CF are using more carbs as fuel source, it will be harmful when they receive high fat diet. Panel E of Fig.2 OGTT results are surprising how healthy control mice got glucose tolerance? The effect is very minimal, again for a fair comparison a pathological control is needed. Glucose stimulated insulin secretion(GSIS) is missing. Insulin tolerance test, pyruvate tolerance test should be added.

Have you injected CF along with glucose during OGTT?

  1. Plasma insulin levels are needed, and other plasma/serum biochemical parameters could be added.
  2. Fig 3: Not a significance effect observed. Panel B enzyme activity is not from mitochondria, as authors have used tissue extract.
  3. 4: Healthy mouse doesn’t have higher ROS like any disease states. Checking ROS at healthy mice doesn’t add any value without adding pathological control or exercised mouse control. The significant effect authors got at panel B ant-A control has broad expansion from approx. 25-120. It is not a acceptable control. We are not expecting any oxidative stress in healthy mouse so panel  E and F are not giving any additional information.
  4. 5. Effects are very minimal, panel A,B,C of CF group has an outlier, so it is considered to be panel A has a non-significant effect.
  5. 6: Authors didn’t show increased sirt3 effect at the protein level in the first 5 figures, There is no justifiable data available why you have used sirt3 KO mouse.
  6. The authors didn’t mention how many functional mitochondria they found during the mito functional assays, and their copy number. There are no tissue histology information provided.
  7. Authors have vast discussed about their results in sections, many of them are not fitting in with their results for example, mitochondria biogenesis. It is not present in your results, the PGC1a mRNA level in table 3 is non-significant effect. Discussion is not attributing to the present results.

Author Response

The authors would like to thank the reviewers for taking the time to carefully evaluate our manuscript and the constructive feedback they have provided. We believe that this feedback has allowed us to greatly improve the clarity and quality of our manuscript. Kindly refer to the itemized point-by-point response.

Reviewer #2

Comment 1:

The authors have Introduced epicatechin as a major component of cocoa powder, as per table 1. Theobromine is a major component (7.12%). They could have mention cocoa powder extract information ie., components in extraction, how you perform the extraction from cocoa powder. And how you decided the dose, 302.1mg/kg/B.W?

Response:

We agree that theobromine is quantitatively a major component of cocoa powder. However, to the best of our knowledge, no effect of theobromine on mitochondria have been yet reported. Moreover, to avoid any specific influence of theobromine on mitochondrial metabolism, the control mice received a vehicle composed of similar content of theobromine and caffeine also dissolved in carboxymethylcellulose. All this information is described in the manuscript (l.93-101). Although supraphysiological concentrations (10-100µM) have been used, clear evidence of positive effects has been observed using (-)-epicatechin concentrations in the low molecular range (as low as 1µM), which is close to the concentrations encountered in vivo (Chidambaram et al. 2020). We are currently using enriched cocoa in human studies with positive effects, we therefore decided to use similar dose in mice. We use guidance for industry providing common conversion factors for deriving a human equivalent dose in mice (Alert et a. 2005).

Comment 2:

15-days of supplementation on the healthy mouse is not a good choice of model to see the effect. Why you have not studied the effect on the high-fat diet-fed mouse, where glucose tolerance is a major problem. Also, long-term treatment for 30-60 days may give significant changes.

Response:

The aim of the study is to determine whether or not cocoa flavanols are able to enhance mitochondrial function and to determine if the sirtuins pathway is involved in the response. Mitochondria is involved in maintaining normal and healthy physiology. They are the main source of ATP in the cell and play a pivotal role in cell life and cell death. Mitochondrial function lowers over aging even in healthy subjects (Gouspillou et al., 2014; Tonkonogi et al., 2003). Therefore, enhancement of mitochondrial function would also be interesting in healthy subjects as it may improve their aerobic capacity and their quality of life. Moreover, we did not choose a model of mitochondrial dysfunction to avoid any interference between the pathology and the cocoa flavanol supplementation.

The supplementation has been determined based on previous studies that explored the effects of (-)-epicatechin on mitochondria (Nogueira et al., 2011; Panneerselvam et al., 2013; Watanabe et al., 2014). Moreover, in previous studies in mice, treatment ranged from 10 to 15 days (Moreno-Ulloa et al., 2015; Nogueira et al., 2011; Panneerselvam et al., 2013; Ramirez-Sanchez et al., 2014; Ramírez-Sánchez et al., 2016; Watanabe et al., 2014). Therefore, we decided to test a 15-day treatment in line with methodology used in previous studies.

Measurement of whole-body metabolism and glucose metabolism should be considered as complementary measures to determine if mitochondrial adaptations would impact whole body metabolism. The manuscript has been modified to point out that mitochondria is the primary outcome (l429-431). We also modified the title to focus on mitochondria and whole-body metabolism rather than on glucose metabolism.

Comment 3:

Article title to results is not matching, the title is very general. But you have only studied gastrocnemius and soleus muscles. The major metabolic organs are not involved in this study like liver, pancreas.

Response:

The title has been modified to indicate that the mitochondrial function was assessed in skeletal muscle. Please read the title as follow: “Dietary cocoa flavanols enhance mitochondrial function in skeletal muscle and modify whole-body metabolism in healthy mice”

Comment 4:

All the enzyme activity you studied is on frozen tissues samples, It is a total tissue protein not mitochondria alone.

Response:

We totally agree, the measurements were made on whole muscle and not on isolated mitochondria. The oxidative phosphorylation system consists of five multimeric complexes embedded in the inner mitochondrial membrane. The assessment of the oxidative phosphorylation enzymes have been developed in cells, isolated mitochondria and tissues (Barrientos et al., 2009). To ensure that activities measured is not influenced by non-mitochondrial enzymes, electron transport chain inhibitors were used (rotenone for the complex I assay, malonate for the complex II assay and KCN for the complex IV assay). These protocols are widely used to determine mitochondrial enzyme activity in whole frozen tissue (Jarreta et al., 2000).

Comment 5:

1: the effect of CF supplementation is very minimal with a broad range of differences between mice. There is no pathological (either diabetes or high-fat diet-fed mice) control

Response:

The significant effects observed with CF supplementation ranged from 6 to 15% which could be considered as significant especially in the context of healthy animals.

Please refer to the comment 2 about the choice to study the effect of cocoa flavanols supplementation in healthy mice.

Comment 6:

2: Panel A. RER below 0.7 is calories from fat. Above 0.7 is carbohydrate. Both control and CF supplement groups using carbs as their fuel source. According to this result mouse supplemented with CF are using more carbs as fuel source, it will be harmful when they receive high fat diet. Panel E of Fig.2 OGTT results are surprising how healthy control mice got glucose tolerance? The effect is very minimal, again for a fair comparison a pathological control is needed. Glucose stimulated insulin secretion (GSIS) is missing. Insulin tolerance test, pyruvate tolerance test should be added.

Response:

Glucose homeostasis is maintained through a multiple interacting complex feedback system involving hormones (Top et al., 2017). The major hormone associated with glycemic control is insulin. High fat diet in animal produces obese animals that have been characterized by a resistance to insulin-stimulated glucose uptake (Nagy and Einwallner, 2018). There is therefore an interest to test CF supplementation to alleviate insulin-resistance induced by HF diet. In line with this hypothesis, Ramirez-el al. observed that epicatechin supplementation reverts HF diet-induced changes of blood glucose levels (Ramírez-Sánchez et al., 2016). While beneficial effects of polyphenol consumption (from green tea, mangrove) have been observed on glucose uptake, the effect of cocoa flavanols remains to be studied.

Several considerations should be taken in account to explore glucose metabolism. Indeed, apart from the factors inherent to the mouse model, the choice of the test is of interest. The first screening test is to measure insulin and glucose. To further characterize the metabolic phenotype an oral glucose tolerance test and/or an insulin tolerance tests can be performed (Ayala et al., 2010). Insulin level measurement would help to explain differences in glucose tolerance. Further tests such as hyperglycemic clamp or hyperinsulemic-euglycemic test should be conducted to better identify the mechanism underpinning the improved glucose tolerance test. In our study, measurement of glucose metabolism should be considered as a complementary measure to determine whether mitochondrial adaptations would impact whole body metabolism or not. The aim was not to explore the influence of cocoa flavanols on glucose metabolism itself as it has been widely studied (for review see: (Lin et al., 2016)). The title has been modified to highlight the effect on whole-body metabolism. Please read as follow: “Dietary cocoa flavanols enhance mitochondrial function in skeletal muscle and modify whole-body metabolism in healthy mice”. Moreover, we do not have plasma sample left to measure insulin. It will require to do a new batch of experiment.

Comment 7:

Have you injected CF along with glucose during OGTT?

Response:

Thank you for your comment, we added the following information in the method section (2.5. Oral glucose tolerance test): “The measurement was performed on the last day of CF supplementation. »

Comment 8:

Plasma insulin levels are needed, and other plasma/serum biochemical parameters could be added.

Response:

Please refer to the second part of the response to the comment 6.

Comment 9:

Fig 3: Not a significance effect observed. Panel B enzyme activity is not from mitochondria, as authors have used tissue extract.

Response:

We did not clearly identify what is the expectation about the significance effect observed. The results of panel B reflect mitochondrial activity from whole muscle tissue as discussed in the comment 4.

Comment 10:

4: Healthy mouse doesn’t have higher ROS like any disease states. Checking ROS at healthy mice doesn’t add any value without adding pathological control or exercised mouse control. The significant effect authors got at panel B ant-A control has broad expansion from approx. 25-120. It is not a acceptable control. We are not expecting any oxidative stress in healthy mouse so panel  E and F are not giving any additional information.

Response:

The aim of the study is to explore mitochondrial function that refers to ATP production, ROS production and Calcium metabolism handling. The permeabilized fiber technique reflects the maximal capacity of mitochondria to produce ROS or consume oxygen. Our results suggest that coca flavanols supplementation lowers mitochondrial ROS production express per oxphos capacity (Figure 4.D). In order to identify an effect of CF supplementation on cellular oxidative stress as it could be hypothesized from the latter results, we explored SOD2 content and total protein carbonylation. Nevertheless, the results did not show any effect of CF supplementation.

Comment 11:

  1. Effects are very minimal, panel A,B,C of CF group has an outlier, so it is considered to be panel A has a non-significant effect.

Response:

We removed the outlier value in panel A and C. The statistical analysis still reveals a difference for NAD+ content and the NAD+/NADH ratio with a p-value increased from 0.084 to 0.092 which could be considered as a trend. As there is no difference, we did not remove the outlier point on the graph. Please find the result of statistical analysis below:

Comment 12:

6: Authors didn’t show increased sirt3 effect at the protein level in the first 5 figures, There is no justifiable data available why you have used sirt3 KO mouse.

Response:

We agree that we did not explore the Sirt3 protein content. However, we observed an effect of CF supplementation on NAD metabolism. Sirtuins were key metabolic sensors that use intracellular metabolites such as NAD to modulate mitochondrial turnover and function (Carafa et al., 2016). We assumed that the improvement of NAD metabolism would simulate Sirtuin activity and therefore modulate mitochondrial function. Moreover Aragones et al. (Aragonès et al., 2016) showed that (-)-epicatchin supplementation caused an increase in NAD content resulting in Sirt1 stimulation. Considering that Sirt 3 modulates mitochondrial function (Lin et al., 2014), we assumed that positive effect of CF supplementation on mitochondrial function would be, at least, partially blunted in Sirt3 KO mice if Sirt3 is involved in CF supplementation adaptation. We do not believe that the assessment of Sirt3 content or activity will be informative in this context. We added the following sentence (l92-94) to clearly explain the use of the Sirt3-/- mice in our study: “The Sirt3-/- mice were used to explore the involvement of Sirt3 in CF-induced mitochondrial biogenesis and whole-body metabolism adaptations. »

Comment 13:

The authors didn’t mention how many functional mitochondria they found during the mito functional assays, and their copy number. There are no tissue histology information provided.

Response:

We used permeabilized fiber technique to analyze functional mitochondria in situ as previously described (Kuznetsov et al., 2008). This protocol allows to characterize mitochondrial function in their normal intracellular position and assemble, preserving essential interaction with other organelles. The results were expressed per muscle weight and are well correlated with mitochondrial content determined by electron microscopy (r=0.92) which is the gold standard measurement (Larsen et al., 2012). We assume that our results on mitochondrial density are robust and that mitochondrial histological assessment would not add complementary results.

Comment 14:

Authors have vast discussed about their results in sections, many of them are not fitting in with their results for example, mitochondria biogenesis. It is not present in your results, the PGC1a mRNA level in table 3 is non-significant effect. Discussion is not attributing to the present results.

Response:

Differential mRNA expression studies implicitly assume that changes in mRNA expression have biological meaning, most likely mediated by corresponding changes in protein levels. Yet, studies into mRNA-protein correspondence have shown notoriously poor correlation between mRNA and protein expression levels, creating concern for inferences from only mRNA expression data (de Sousa Abreu et al., 2009). The relationship between protein and mRNA expression levels informs about the combined outcomes of translation and protein degradation which are, in addition to transcription and mRNA stability, essential contributors to gene expression regulation. Therefore, a similar mRNA level pre vs. post supplementation did not mean that there is no difference in protein content or in protein activity. Moreover, a critical function of proteins is their activity as enzymes, which are needed to catalyze biological reaction. The regulation of their function allows the cell to regulate not only the amounts but also the activity of its protein constituents (Cooper, 2000). These regulations include phosphorylation, ubiquinitation, nitrosylation, glycosylation,… and we may not rule out that CF supplementation will stimulate some of these mechanisms. We acknowledge that discussing this point would be of interest. However including a specific part in the discussion will lengthen the manuscript and increase the number of references that we have been asked to reduce.

References:

Aragonès, G., Suárez, M., Ardid-Ruiz, A., Vinaixa, M., Rodríguez, M.A., Correig, X., Arola, L., Bladé, C., 2016. Dietary proanthocyanidins boost hepatic NAD(+) metabolism and SIRT1 expression and activity in a dose-dependent manner in healthy rats. Sci Rep 6, 24977. https://doi.org/10.1038/srep24977

Ayala, J.E., Samuel, V.T., Morton, G.J., Obici, S., Croniger, C.M., Shulman, G.I., Wasserman, D.H., McGuinness, O.P., 2010. Standard operating procedures for describing and performing metabolic tests of glucose homeostasis in mice. Dis Model Mech 3, 525–534. https://doi.org/10.1242/dmm.006239

Barrientos, A., Fontanesi, F., Díaz, F., 2009. Evaluation of the Mitochondrial Respiratory Chain and Oxidative Phosphorylation System using Polarography and Spectrophotometric Enzyme Assays. Curr Protoc Hum Genet CHAPTER, Unit19.3. https://doi.org/10.1002/0471142905.hg1903s63

Carafa, V., Rotili, D., Forgione, M., Cuomo, F., Serretiello, E., Hailu, G.S., Jarho, E., Lahtela-Kakkonen, M., Mai, A., Altucci, L., 2016. Sirtuin functions and modulation: from chemistry to the clinic. Clin Epigenetics 8, 61. https://doi.org/10.1186/s13148-016-0224-3

Cooper, G.M., 2000. Regulation of Protein Function. The Cell: A Molecular Approach. 2nd edition.

de Sousa Abreu, R., Penalva, L.O., Marcotte, E.M., Vogel, C., 2009. Global signatures of protein and mRNA expression levels. Mol. BioSyst. 10.1039.b908315d. https://doi.org/10.1039/b908315d

Gouspillou, G., Bourdel-Marchasson, I., Rouland, R., Calmettes, G., Biran, M., Deschodt-Arsac, V., Miraux, S., Thiaudiere, E., Pasdois, P., Detaille, D., Franconi, J.-M., Babot, M., Trézéguet, V., Arsac, L., Diolez, P., 2014. Mitochondrial energetics is impaired in vivo in aged skeletal muscle. Aging Cell 13, 39–48. https://doi.org/10.1111/acel.12147

Jarreta, D., Orús, J., Barrientos, A., Miró, O., Roig, E., Heras, M., Moraes, C.T., Cardellach, F., Casademont, J., 2000. Mitochondrial function in heart muscle from patients with idiopathic dilated cardiomyopathy. Cardiovasc. Res. 45, 860–865. https://doi.org/10.1016/s0008-6363(99)00388-0

Kuznetsov, A.V., Veksler, V., Gellerich, F.N., Saks, V., Margreiter, R., Kunz, W.S., 2008. Analysis of mitochondrial function in situ in permeabilized muscle fibers, tissues and cells. Nat Protoc 3, 965–976. https://doi.org/10.1038/nprot.2008.61

Larsen, S., Nielsen, J., Hansen, C.N., Nielsen, L.B., Wibrand, F., Stride, N., Schroder, H.D., Boushel, R., Helge, J.W., Dela, F., Hey-Mogensen, M., 2012. Biomarkers of mitochondrial content in skeletal muscle of healthy young human subjects. J Physiol 590, 3349–3360. https://doi.org/10.1113/jphysiol.2012.230185

Lin, L., Chen, K., Khalek, W.A., Ward, J.L., Yang, H., Chabi, B., Wrutniak-Cabello, C., Tong, Q., 2014. Regulation of Skeletal Muscle Oxidative Capacity and Muscle Mass by SIRT3. PLoS One 9, e85636. https://doi.org/10.1371/journal.pone.0085636

Lin, X., Zhang, I., Li, A., Manson, J.E., Sesso, H.D., Wang, L., Liu, S., 2016. Cocoa Flavanol Intake and Biomarkers for Cardiometabolic Health: A Systematic Review and Meta-Analysis of Randomized Controlled Trials. The Journal of Nutrition 146, 2325–2333. https://doi.org/10.3945/jn.116.237644

Moreno-Ulloa, A., Nogueira, L., Rodriguez, A., Barboza, J., Hogan, M.C., Ceballos, G., Villarreal, F., Ramirez-Sanchez, I., 2015. Recovery of Indicators of Mitochondrial Biogenesis, Oxidative Stress, and Aging With (-)-Epicatechin in Senile Mice. J. Gerontol. A Biol. Sci. Med. Sci. 70, 1370–1378. https://doi.org/10.1093/gerona/glu131

Nagy, C., Einwallner, E., 2018. Study of In Vivo Glucose Metabolism in High-fat Diet-fed Mice Using Oral Glucose Tolerance Test (OGTT) and Insulin Tolerance Test (ITT). JoVE (Journal of Visualized Experiments) e56672. https://doi.org/10.3791/56672

Nogueira, L., Ramirez-Sanchez, I., Perkins, G.A., Murphy, A., Taub, P.R., Ceballos, G., Villarreal, F.J., Hogan, M.C., Malek, M.H., 2011. (-)-Epicatechin enhances fatigue resistance and oxidative capacity in mouse muscle. J. Physiol. (Lond.) 589, 4615–4631. https://doi.org/10.1113/jphysiol.2011.209924

Panneerselvam, M., Ali, S.S., Finley, J.C., Kellerhals, S.E., Migita, M.Y., Head, B.P., Patel, P.M., Roth, D.M., Patel, H.H., 2013. Epicatechin regulation of mitochondrial structure and function is opioid receptor dependent. Mol Nutr Food Res 57, 1007–1014. https://doi.org/10.1002/mnfr.201300026

Ramirez-Sanchez, I., De los Santos, S., Gonzalez-Basurto, S., Canto, P., Mendoza-Lorenzo, P., Palma-Flores, C., Ceballos-Reyes, G., Villarreal, F., Zentella-Dehesa, A., Coral-Vazquez, R., 2014. (-)-Epicatechin improves mitochondrial-related protein levels and ameliorates oxidative stress in dystrophic δ-sarcoglycan null mouse striated muscle. FEBS J. 281, 5567–5580. https://doi.org/10.1111/febs.13098

Ramírez-Sánchez, I., Rodríguez, A., Moreno-Ulloa, A., Ceballos, G., Villarreal, F., 2016. (-)-Epicatechin-induced recovery of mitochondria from simulated diabetes: Potential role of endothelial nitric oxide synthase. Diab Vasc Dis Res 13, 201–210. https://doi.org/10.1177/1479164115620982

Tonkonogi, M., Fernström, M., Walsh, B., Ji, L.L., Rooyackers, O., Hammarqvist, F., Wernerman, J., Sahlin, K., 2003. Reduced oxidative power but unchanged antioxidative capacity in skeletal muscle from aged humans. Pflugers Arch. 446, 261–269. https://doi.org/10.1007/s00424-003-1044-9

Top, M. van den, Zhao, F.-Y., Viriyapong, R., Michael, N.J., Munder, A.C., Pryor, J.T., Renaud, L.P., Spanswick, D., 2017. The impact of ageing, fasting and high-fat diet on central and peripheral glucose tolerance and glucose-sensing neural networks in the arcuate nucleus. Journal of Neuroendocrinology 29, e12528. https://doi.org/10.1111/jne.12528

Watanabe, N., Inagawa, K., Shibata, M., Osakabe, N., 2014. Flavan-3-ol fraction from cocoa powder promotes mitochondrial biogenesis in skeletal muscle in mice. Lipids Health Dis 13, 64. https://doi.org/10.1186/1476-511X-13-64

Reviewer 3 Report

The paper from Daussin et al. is aimed to evaluate the ability of  dietary supplementation with cocoa flavanols (CF) to enhance mitochondrial functions in different muscles  of wild type mice. The authors  studied respiratory functions, mitochondrial mass, ROS production and NAD metabolism,  along with the possible role of mitochondrial Sirt3. While the topic is interesting, given  the growing attention the scientific community is given to natural compounds able to improve mitochondrial function, I have some concerns regarding this paper. It appears that some of the conclusions  reported are mainly supported from the literature that the authors cite (indeed 78 references are too much for a research article) than from their experimental results.

My first concern regards the effect of CF supplementation on mitochondrial super-complexes. The authors  says: “Visual analysis suggests…” (line 343-results section and line 506-discussion section).  A visual analysis cannot be used to support a scientific  conclusion. It is only a subjective observation that simply doesn’t scientifically sounds. If the authors want to consider this datum, it must be supported by , for example, a densitometric analysis of their results.

Besides, the only appropriate conclusion that can be made in the case of a p-value greater  than 5% is that there was no significant effect. So the instances concerning  a trend should be removed.

The authors show that CF supplementation is able to improve whole-body metabolism, mitochondrial respiration and enzyme activities, along with ROS and NAD metabolisms. They claim a role for Sirt3 in the observed improvements. However, it appears  from their results that CF supplementation does not  induce significant differences in Sirt3 mRNA levels between wild type mice supplemented or not.  And what about the protein levels? The only mRNA level that changes significantly is NRF1, but the authors do not discuss this feature.

The authors provide  the analysis of Sirt3-/- mice supplemented with CF or not as a support  for their conclusion. From their results it appears the some of the changes observed in wild type mice are lost in the knock- out mice.  However,  it would be interesting to first of all evaluate whether  there are significant differences between wild type and knock-out mice.  In my opinion, in order to dissect the specific role of Sirt3 in the considered parameters, the authors should compare the four groups of mice: wt, CF-supplemented wt, Sirt3-/-, and CF-supplemented Sirt3-/-.

Minor questions:

To improve the readability and comprehension of the paper the authors must present their results in the text in the same order as in the figures  (see their comments to  figure 1 as an example).

Figure 2 C. PRCF was evaluated over 24 or 48 hours?

Lines 333 and 338: Fig. 3C ???

Lines 339-357: first the authors must conclude their  comments to figure 3, and only  then can they present the results related to mRNA assessment (in addition, the authors must change “Table 2” at line 342 with “Table 3”).

Figure 4: Why don’t the authors normalize using beta-actin or GAPDH levels as internal standard? I am perplexed with regard to the use of total protein levels.  Not all the proteins  are constitutively expressed.

There is a mistake:  Panel G is totally lacking in the figure.

Author Response

The authors would like to thank the reviewers for taking the time to carefully evaluate our manuscript and the constructive feedback they have provided. We believe that this feedback has allowed us to greatly improve the clarity and quality of our manuscript. Kindly refer to the itemized point-by-point response.

Reviewer #3

Comment 1:

My first concern regards the effect of CF supplementation on mitochondrial super-complexes. The authors  says: “Visual analysis suggests…” (line 343-results section and line 506-discussion section).  A visual analysis cannot be used to support a scientific  conclusion. It is only a subjective observation that simply doesn’t scientifically sounds. If the authors want to consider this datum, it must be supported by, for example, a densitometric analysis of their results.

Response:

Thank you for your comment. To present objective results, we have quantified by densitometric analysis the results. We modified the graph and we added the individual value of each band for each animal. The number of animals was limited and we were not able to perform statistical analysis. We presented the result based on ratio between CF and WT groups as follow: “The densitometric analysis suggests that CF supplementation increased overall supercomplexes content without qualitative adaptation or rearrangements. Indeed, respirasome density was improve by 58% using complex I and complex II probes and by 93% using complex III probe. Similarly, complex I and complex III content embedded in supercomplexes were higher in CF group.” The figure has also been modified accordingly.

Comment 2:

Besides, the only appropriate conclusion that can be made in the case of a p-value greater  than 5% is that there was no significant effect. So the instances concerning a trend should be removed.

Response:

The word trend was removed accordingly.

Comment 3:

The authors show that CF supplementation is able to improve whole-body metabolism, mitochondrial respiration and enzyme activities, along with ROS and NAD metabolisms. They claim a role for Sirt3 in the observed improvements. However, it appears  from their results that CF supplementation does not  induce significant differences in Sirt3 mRNA levels between wild type mice supplemented or not.  And what about the protein levels? The only mRNA level that changes significantly is NRF1, but the authors do not discuss this feature.

Response:

Differential mRNA expression studies implicitly assume that changes in mRNA expression have biological meaning, most likely mediated by corresponding changes in protein levels. Yet studies into mRNA-protein correspondence have shown notoriously poor correlation between mRNA and protein expression levels, creating concern for inferences from only mRNA expression data (de Sousa Abreu et al., 2009). The relationship between protein and mRNA expression levels informs about the combined outcomes of translation and protein degradation which are, in addition to transcription and mRNA stability, essential contributors to gene expression regulation. Therefore, a similar mRNA levels pre vs. post supplementation did not mean that there is no protein content difference. Moreover, a critical function of proteins is their activity as enzymes, which are needed to catalyze biological reaction. The regulation of their function allows the cell to regulate not only the amounts but also the activities of its protein constituents (Cooper, 2000). These regulations include phosphorylation, ubiquinitation, nitrosylation, glycosylation,… and we may not rule out that CF supplementation will stimulate some of these mechanisms. Morevover, we aimed to explore the effect of CF supplementation on functional parameters (mitochondrial respiration, mitochondrial H2O2 production, mitochondrial calcium uptake, whole body metabolism). Although protein content and activity are interesting, they do not modify the general message of our study. We acknowledge that discussing this point would be of interest. However including a specific part in the discussion will lengthen the manuscript and increase the number of references that we have been asked to reduce.

Reference

Cooper, G.M., 2000. Regulation of Protein Function. The Cell: A Molecular Approach. 2nd edition.

de Sousa Abreu, R., Penalva, L.O., Marcotte, E.M., Vogel, C., 2009. Global signatures of protein and mRNA expression levels. Mol. BioSyst. 10.1039.b908315d. https://doi.org/10.1039/b908315d

Comment 4:

The authors provide  the analysis of Sirt3-/- mice supplemented with CF or not as a support  for their conclusion. From their results it appears the some of the changes observed in wild type mice are lost in the knock- out mice.  However,  it would be interesting to first of all evaluate whether  there are significant differences between wild type and knock-out mice.  In my opinion, in order to dissect the specific role of Sirt3 in the considered parameters, the authors should compare the four groups of mice: wt, CF-supplemented wt, Sirt3-/-, and CF-supplemented Sirt3-/-.

Response:

Thank you for your comment, we built a new figure 6 including the four groups of mice. The results and discussion were modified accordingly to take in account this modification.

Comment 5:

To improve the readability and comprehension of the paper the authors must present their results in the text in the same order as in the figures  (see their comments to  figure 1 as an example).

Response:

Thank you for your comment, the figures and/or the text were modified to present the results in a same order in the text and in the figure.

Comment 6:

Figure 2 C. PRCF was evaluated over 24 or 48 hours?

Response:

The PCRF was evaluated over a 24-h period. The information was added in the manuscript as follows: « Percent relative cumulative frequency (PCRF) of RER was determined over a 24-h period as previously described by Riachi et al. (2004). »

Comment 7:

Lines 333 and 338: Fig. 3C ???

Response:

The figure was modified and the text was adapted to refer properly to the figure panels.

Comment 8:

Lines 339-357: first the authors must conclude their  comments to figure 3, and only  then can they present the results related to mRNA assessment (in addition, the authors must change “Table 2” at line 342 with “Table 3”).

Response:

The manuscript was modified accordingly.

Comment 9:

Figure 4: Why don’t the authors normalize using beta-actin or GAPDH levels as internal standard? I am perplexed with regard to the use of total protein levels.  Not all the proteins are constitutively expressed.

Response:

Your remark is completely relevant in the Western Blot analysis. We chose to use Stain-Free total protein measurement as the loading control. Stain-Free imaging allows for the complete elimination of the inherently problematic use of housekeeping proteins as loading controls on western blots. Stain-Free total protein measurement serves as a more reliable loading control than housekeeping proteins like beta-actin or GAPDH (Taylor and Posch, 2014).

Figure 2: Linearity comparison of stain-free total protein measurement and immunodetection of three housekeeping proteins in 10–50 ?g of HeLa cell lysate. On the left are representative images of (a), stain-free blot and the chemi blots for (b), ?-actin; (c), ?-tubulin and (d), GAPDH. Lane labels correspond to total protein load (?g). Although the actin and tubulin signals appear linear, the densitometric ratio was far below the predicted “quantitative response” of actual loading whereas the stain-free signal correlated to the expected result (e) (Taylor and Posch, 2014, https://doi.org/10.1155/2014/361590).

Furthermore, total protein normalization takes into consideration the intensity of all proteins in the lane, sample loading variations in each lane and experimental variations during Western Blot step (electrophoresis, transfer and revelation).

Comment 10:

There is a mistake:  Panel G is totally lacking in the figure.

Response:

The figure legend was modified accordingly.

Round 2

Reviewer 2 Report

The authors Daussin FN eta al, have improved the manuscript with required changes.

I agree that new animal study may not be possible now to work on blood samples and ITT. 

Reviewer 3 Report

The revised version of the manuscript is suitable for the publication.